

# Astroclimes - measuring the abundance of CO2 and CH4 in the Earth's atmosphere using astronomical observations

Marcelo Aron Fetzner Keniger[1,2], David Armstrong[1,2], Matteo Brogi[3,4], Siddharth Gandhi[1,2], and Marina Lafarga[1,2]

[1]Department of Physics, University of Warwick, Gibbet Hill Road, Coventry CV4 7AL, UK
[2]Centre for Exoplanets and Habitability, University of Warwick, Gibbet Hill Road, Coventry CV4 7AL, UK
[3]Dipartimento di Fisica, Università degli Studi di Torino, via Pietro Giuria 1, I-10125, Torino, Italy
[4]INAF-Osservatorio Astrofisico di Torino, Via Osservatorio 20, I-10025 Pino Torinese, Italy

**Correspondence:** Marcelo Aron Fetzner Keniger (Marcelo-Aron.Fetzner-Keniger@warwick.ac.uk)

**Abstract.** Monitoring the abundance of greenhouse gases is necessary to quantify their impact on global warming and climate change. Carbon dioxide (CO2) and methane (CH4) are the two most important greenhouse gases when it comes to global warming, and there are many ground-based networks, such as TCCON, and satellites, such as OCO-2, OCO-3 and GOSAT-2, that are tasked with measuring the total column volume mixing ratio (VMR) of either one or both of these gases. However, these networks all rely on sunlight to carry out their measurements. For column measurements at night, a technique called integrated-path differential absorption (IPDA) has been employed recently using a lidar system. We present a new algorithm, Astroclimes, that hopes to complement and extend nighttime CO2 and CH4 column measurements. Astroclimes can measure the abundance of greenhouse gases on Earth by generating a model telluric transmission spectra and fitting it to the spectra of telluric standard stars in the near-infrared taken by ground-based telescopes. We carried out new observations for one night with the CARMENES spectrograph in the Calar Alto Observatory, Spain, as well as a weather balloon launch to measure a local atmospheric profile. After correcting for a small bias in CO2 estimates, we show that our CO2 and CH4 measurements exhibit good agreement with the refereed literature, and our average relative uncertainties for the column-averaged dry air mole fraction of CO2 and CH4 are 0.4% and 0.5%, respectively. These uncertainties are precision errors based on the 68% confidence intervals of our MCMC analysis posterior distribution, they do not include any systematic errors or biases. A historical analysis of archival data from several different instruments will be carried out in future work to further test our algorithm and to identify and quantify potential systematic biases.

## 1 Introduction

Climate change is one of the most pressing matters in today's society, having already caused substantial damage to ecosystems and leaving a significant portion of the population highly vulnerable to climatic hazards, according to the latest report from the Intergovernmental Panel on Climate Change (IPCC; Lee et al., 2023). Even though the vast majority of climate scientists agrees that humans are responsible for the increase in global temperature that drives climate change (Cook et al., 2016), current policies are still not enough to decrease the emission of greenhouse gases (GHG), which are the cause of this rise in temperature



(Peters et al., 2019; Lee et al., 2023). Among the GHGs, the ones that dominate global warming are carbon dioxide (CO2) and methane (CH4; Lee et al., 2023). As such, obtaining accurate and reliable measurements for the abundance of said gases in the atmosphere is paramount to understanding and modelling climate change, as well as to guide governmental policies for alleviating its impact (Bruhwiler et al., 2021).

Ground-based carbon dioxide measurements are usually taken using a non-dispersive infrared analyser (NDIR; see Komhyr et al. 1989 or Hodgkinson and Tatam 2013; Hodgkinson et al. 2013; Jha 2022 for reviews), but another technique called Cavity Ring-Down Spectroscopy (CRDS; Wheeler et al., 1998; Berden et al., 2000) has also been employed. The basic principle behind the two methods is similar and consists in analysing the transmittance of an infrared (IR) radiation source, such as an LED (Jha, 2022) or a laser (Wheeler et al., 1998), as it goes through a chamber with a sample gas. Each gas molecule will affect the transmittance in a different way, so they can be distinctively identified by analysing the absorption characteristics (Jha, 2022). In the NDIR method, the magnitude of absorption is used to infer the abundance of gas present, whereas the CRDS method is based on the rate of absorption instead (Berden et al., 2000).

The longest record of direct measurements of CO2 in the atmosphere comes from the Mauna Loa Observatory in Hawaii, with ongoing observations since the 1950s (Pales and Keeling, 1965; Keeling et al., 1976; Bacastow et al., 1985; Komhyr et al., 1989; Thoning et al., 1989). At first, these measurements were carried out solely by the Scripps Institution of Oceanography (SIO; Pales and Keeling, 1965), but since 1974 the Global Monitoring Laboratory (GML)[1], previously known as the Geophysical Monitoring for Climate Change (GMCC) program, has also been carrying out continuous CO2 measurements in Mauna Loa (Komhyr et al., 1989; Thoning et al., 1989). The GML is a global network that monitors atmospheric parameters such as the amount of CO2 and other GHGs. Besides Mauna Loa, there are three other baseline observatories (American Samoa, Alaska and Antarctica), in addition to measurements from tall towers (Bakwin et al., 1998; Peters et al., 2007; Andrews et al., 2014), small aircraft (Sweeney et al., 2015), weather balloons (Karion et al., 2010) and more than 50 sites spread worldwide (Conway et al., 1994; Ballantyne et al., 2012)[2]. Initially, the GML employed the NDIR technique to measure CO2 abundances in Mauna Loa, but since 2019 they switched to a detector that uses the CRDS technique[3].

Since the start of the continuous observations of atmospheric CO2 in Mauna Loa, certain trends have become evident: a seasonal variation and a long term increase (Pales and Keeling, 1965; Keeling et al., 1976; Komhyr et al., 1989; Thoning et al., 1989). The seasonal variation is caused by the uptake and release of CO2 by the land biosphere (Junge and Czeplak, 1968). The long term increase has been attributed to anthropogenic global emissions of CO2 (Watts, 1980; Keeling et al., 1985; Thoning et al., 1989) which are dominated by the combustion of fossil fuels (Peters et al., 2019; Lee et al., 2023). This increase in CO2 in the atmosphere is a direct cause of the observed rise in global temperature (Hansen et al., 2010; Voosen, 2021), which had been predicted by a number of climate models (Watts, 1980; Cox et al., 2000; Friedlingstein et al., 2006). To keep on monitoring the state of the climate and to assess the reliability of climate models, it is crucial that continuous atmospheric CO2 measurements are maintained.

---

[1] https://gml.noaa.gov/

[2] https://gml.noaa.gov/ccgg/about.html

[3] https://gml.noaa.gov/ccgg/about/co2_measurements.html



The CO2 measurements from the four GML Baseline Observatories have the advantage of being taken in remote locations in different parts of the globe, so they accurately represent the background atmosphere and serve as the backbone of the GML's climate monitoring (Stanitski et al., 2018). However, since these measurements are taken at ground level, they have limited information regarding the total column abundance of CO2 in the atmosphere, which adds uncertainty to the study of the growth rate of atmospheric CO2 (Friedlingstein et al., 2023). The aforementioned aircraft measurements have the capability of doing

so, but they only go up to ∼ 13km and are taken mostly inside the United States[4].

For column measurements of CO2, satellites such as NASA's Orbiting Carbon Observatory 2 (OCO-2; Basilio et al., 2014; Crisp et al., 2017) and, more recently, OCO-3 (Eldering et al., 2019) were launched, as well as the Greenhouse gases Observing SATellite 2 (GOSAT-2), by the Japanese Aerospace Exploration Agency (JAXA; Imasu et al., 2023), which also includes column measurements of CH4. These satellites observe sunlight reflected on the Earth's surface and measure the CO2 abundance

by analysing the solar spectra in the near-infrared (NIR). When light passes through the Earth's atmosphere, it is left with an imprint based on which gases are present in the air. This imprint comes in the form of absorption lines, and the absorption lines left by molecules in the Earth's atmosphere are called telluric lines. Each gas absorbs light at specific wavelengths, so their spectral lines can be identified and by measuring the intensity of their spectral lines, the abundance of said gas can be obtained.

There are also ground-based networks focused on measuring the column abundances of relevant greenhouse gases such

as CO2 and CH4, one of them being the Total Carbon Column Observing Network (TCCON; Wunch et al., 2011), whose measurements are directly comparable to those from space-based instruments and thus can provide a link between satellite and ground-based measurements (Wunch et al., 2011).

The TCCON measurements, however, still depend on sunlight. So, to summarise the current state of CO2 measurements, we have: *in-situ* measurements taken during the day and during the night in several stations across the world; satellite measure-

ments of the column abundance that rely on sunlight, thus can only be taken during the day; column abundances measured by a ground-based network, which can link the previous two types of measurements, but also rely on sunlight.

More recently, nighttime measurements of the column abundance of CO2 started being carried out by the Atmospheric Environment Monitoring Satellite (AEMS; Pei et al., 2023) and NASA's Active Sensing of CO2 Emissions over Nights, Days, and Seasons (ASCENDS; Mao et al., 2024). Both missions employ an integrated-path differential absorption (IPDA) lidar

system, which in addition to allowing column measurements at night, also works on cloudy conditions and high latitudes, which the previous networks struggle with.

Here, we describe a novel method for measuring the column abundance of CO2 that aims to complement and extend nighttime measurements. Our measurement approach is similar to that of OCO-2 and TCCON, but instead of using sunlight, we analyse the spectra of stars taken by ground-based telescopes. The selected sample of stars is known as telluric standard

stars (Vacca et al., 2003; Seifahrt et al., 2010), usually O, B or A type stars that are called telluric standards because they are very hot, thus have very few spectral lines, and rotate very fast, thus the few lines that they do have are broadened, making the stellar lines easy to distinguish from the narrower telluric lines (Ulmer-Moll et al., 2019).

[4]https://gml.noaa.gov/ccgg/aircraft/





Telluric standards are a common byproduct of ground-based astronomical spectroscopic observations. Spectroscopic observations are used in astronomy, for example, to detect exoplanets through the radial velocity (RV) method (Mayor and Queloz, 1995; Lovis et al., 2006) and to study exoplanet atmospheres (Brogi et al., 2012, 2013, 2014; Madhusudhan et al., 2014; Crossfield, 2015). Removal of the telluric lines is essential in both cases, but more so for exoplanet atmosphere studies, since the telluric signal can be orders of magnitude higher than the planet signal (Brogi et al., 2014).

Historically, telluric lines have been removed from stellar spectra using the standard star method (Vidal-Madjar et al., 1986; Vacca et al., 2003), which consists in dividing the science spectra by the spectra of a telluric standard star. This technique, however, requires the standard star to be observed close in time and airmass to the science target (Vacca et al., 2003), so that atmospheric conditions are as similar as possible, thus costing telescope time that could otherwise be used for other scientific purposes. Additionally, no matter how void of spectral lines a star's spectra may be, it is never featureless, which will inevitably affect the resulting science spectra (Lallement et al., 1993; Bailey et al., 2007; Ulmer-Moll et al., 2019).

An alternative telluric removal method has been put forward by many authors (Lallement et al., 1993; Bailey et al., 2007; Seifahrt et al., 2010; Cotton et al., 2014) and is known as the synthetic transmission method. As the name suggests, instead of using the spectra of a telluric standard, a model telluric spectra is computed, and the telluric lines are removed by dividing the science spectra by this model telluric spectra. Generating a synthetic telluric spectra usually requires atmospheric parameters such as pressure, temperature and abundances as a function of height, coupled with a molecular line database and a radiative transfer model (Seifahrt et al., 2010; Ulmer-Moll et al., 2019).

There are many tools in the literature made for removing telluric lines using the synthetic transmission method, such as Molecfit (Smette et al., 2015), TelFit (Gullikson et al., 2014) and TAPAS (Bertaux et al., 2014). These three telluric correction codes are analysed and compared between each other and also with the standard star method in Ulmer-Moll et al. (2019), where they conclude that Molecfit is the most complete package between the three, and that synthetic transmission has some advantages over the standard star method when dealing with water lines, but performs worse for oxygen lines.

All of these tools, however, were made primarily to remove the telluric lines, not to study them, so often their fitting approach is a simple Levenberg-Marquardt least-squares fit (Gullikson et al., 2014; Smette et al., 2015) and uncertainties for the molecular abundances are not reported. Thus, we created our own synthetic transmission code, called "Astroclimes". Developing our own code also grants us more control over the modelling process and allows us to tailor its performance to our specific needs, which in this case is to measure the abundance of greenhouse gases in the Earth's atmosphere, setting it apart from the other telluric removal tools in the literature[5].

By utilising telluric standard stars as opposed to the Sun, we can perform column measurements at night, which could provide insights on the natural cycles of the studied molecules. Additionally, since telluric standards are a common byproduct of ground-based astronomical spectroscopic observations, there are plenty of archival data at our disposal in telescopes spread across the world, thus providing a potential new measurement network that could be used to complement current global climate models.

---

[5]The capabilities of our code for removing telluric lines will be tackled in future work



This paper is structured as follows: first, we explain the step-by-step of our new method in Section 2; then, in Section 3, we describe all of the different data sets employed in our analysis and how they were handled; Section 4 contains our results and discussion, where we explore the influence of using different atmospheric profiles and test our model against the ESO Sky Model[6] and against existing observational data; and finally in section 5 we present our conclusions.

## 2 Methodology

### 2.1 Generating the model spectra

The end product of our algorithm is a model telluric spectra that contains absorption lines of certain molecular species present in the Earth's atmosphere. The two main ingredients needed to compute this model spectra are the cross-sections of the desired molecules and an atmospheric profile.

The cross-sections were calculated in a similar fashion as in Gandhi and Madhusudhan (2017) and Gandhi et al. (2020). However, in these papers they calculated a grid of molecular cross-sections for volatile species found in giant planet atmospheres in a "high temperature" regime, which goes from 300-3500 K. The Earth's atmosphere is below this regime, so a new grid was calculated specifically for this work for a "low temperature" regime. This grid contains cross-section values for 11 different pressures ranging from 0-5 log(Pa) in 0.5 log(Pa) increments, 11 different temperatures ranging from 100-350 K in 25 K increments, and 2480001 wavelength values ranging from 0.4-50 $\mu$m in constant wavenumber steps of 0.01 cm$^{-1}$. The molecules available are CO2, CH4, H2O, O2, N2, CH3Cl, CO, H2, HCN, N2O, NO2, O3 and OH. To compute all of the cross-sections, we used the line lists from the HITRAN database (Gordon et al., 2022).

The atmospheric profile describes how the pressure $P$, the temperature $T$ and the molecular abundances $x_i$ for each molecule $i$ vary as a function of height. The process of obtaining the atmospheric profile is explained in Section 3.2. The atmospheric profile gives the aforementioned parameters for certain height levels. One atmospheric "layer" is the space between said levels. Currently, the characteristic values for $P$, $T$ and $x_i$ taken are the mid-point of each layer, but there are plans to refine that in future work to take into account the non-linear relation between the variables. From the pressure and the temperature, a value for the cross-section of the desired molecules as a function of wavelength $\lambda$, $\sigma_i(\lambda)$, is linearly interpolated from the cross-sections grid. With that, we can calculate the amount of light that is lost as it goes through the atmospheric layers. This is quantified by the opacity $\tau(\lambda)$, which for molecular line absorption is given by:

$$\tau_z(\lambda) = nr \sum_i x_i \sigma_i(\lambda) \tag{1}$$

where

$$n = \frac{P}{k_B T} \tag{2}$$

---

[6]https://www.eso.org/sci/software/pipelines/skytools/skymodel





is the number density of particles in the atmosphere, with $k_B = 1.380649 \times 10^{-23}$ m$^2$ kg s$^{-2}$ K$^{-1}$ being the Boltzmann

constant, and

$$r = \frac{Z_{\text{top}} - Z_{\text{bottom}}}{\cos \theta} \tag{3}$$

is the distance travelled by the light inside that layer, where $Z_{\text{top}}$ refers to the height at the top limit of the layer, $Z_{\text{bottom}}$ refers to the height at the bottom limit of the layer, and $\theta$ is the angle between the target and the local zenith. $\theta$ is related to the airmass as:

$$\cos \theta = \frac{1}{\text{airmass}} \tag{4}$$

The index $z$ in Equation (1) indicates that the opacity is calculated for each atmospheric layer, starting at the top of the atmosphere $h_{\text{top}}$ and going all the way down to the observatory height $h_0$. The transmission $T(\lambda)$ is then calculated from the sum of the opacity in all layers as expressed below:

$$T(\lambda) = e^{-\sum_{h_{\text{top}}}^{h_0} \tau_z(\lambda)} \tag{5}$$

Apart from line absorption by molecular species in the atmosphere, there are other effects caused by the interaction of light with particles that can change the transmission spectra. Two such effects are described in Noll et al. (2012), where the authors describe an atmospheric radiation model for Cerro Paranal, one of the European Southern Observatory's (ESO)[7] astronomical observatories in Chile. These effects are Rayleigh scattering and aerosol scattering, which can be parameterised by the following expressions, respectively:

$$\tau_R(\lambda) = \frac{P_0}{1013.25} \left( 0.00864 + 6.5 \times 10^{-6} h_0 \right)$$

$$\times \lambda^{-\left( 3.916 + 0.074\lambda + \frac{0.050}{\lambda} \right)} \tag{6}$$

$$T_{\text{aerosol}}(\lambda) = 10^{-0.4 k(\lambda) \times \text{airmass}} \quad , \tag{7}$$

where

$$k(\lambda) = k_0 \lambda^{\alpha} \quad , \tag{8}$$

where $k_0 = 0.013 \pm 0.002$ mag/airmass and $\alpha = -1.38 \pm 0.06$, with the wavelength $\lambda$ in $\mu$m. In Equation (6), $P_0$ is the site pressure in hPa and $h_0$ is the site height in km.

---

[7]https://www.eso.org/public/



Another effect that is particularly important in the region containing the O2 lines is collision-induced absorption (CIA), described in Gordon et al. (2017) and Karman et al. (2019). This type of absorption occurs when two species experience a close encounter and give rise to a transient or interaction-induced dipole moment (Hubeny and Mihalas, 2014). Since this is a second order interaction, its opacity is calculated by:

$$\tau_{\text{CIA}}(\lambda) = (nr)^2 \sum_{i,j} x_i x_j \alpha_{i,j}(\lambda) \quad , \tag{9}$$

where $\alpha_{i,j}(\lambda)$ are the coefficients associated with a certain molecular collision involving molecules $i$ and $j$. These coefficients can be obtained from the HITRAN database[8] (Rothman et al., 2013) for a number of molecular collisions. For this work, we included the absorption caused by the collision of O2 molecules with other atmospheric molecules. The HITRAN database has a special file for this, named "O2-air", which includes O2-O2, O2-N2 and O2-Ar collisions.

The coefficients in these files are measured for a certain wavelength range and a certain temperature, and in some cases the same collision can have different reported coefficients for different temperatures, whereas others only have coefficients for one given temperature value. Whenever there are multiple temperature values, if our atmospheric temperature is inside this range, then the resulting coefficients are obtained by linear interpolation. If the atmospheric temperature is outside the temperature range, we use the coefficient of the closest temperature value.

The final transmission spectra, with all of the aforementioned effects included, is given by:

$$T(\lambda) = e^{-\sum_{h_{\text{top}}}^{h_0} \tau_z(\lambda)} \times e^{-\sum_{h_{\text{top}}}^{h_0} \tau_{\text{CIA}}(\lambda)} \times e^{-\tau_R(\lambda)} \times T_{\text{aerosol}}(\lambda) \tag{10}$$

The model given by Equation (10) has not yet accounted for instrumental broadening, which is a necessary step that has to be done before comparing our model to observations. This is done by the convolution of the unbroadened spectra and a kernel that describes the line spread function, that is, the shape of the spectral lines. We chose a Gaussian kernel to model our lines, and its width depends on the resolution of the instrument where the data comes from. An important step in this process is that we first convert our wavelength distribution to have a constant resolving power $R = \frac{\lambda}{\Delta\lambda}$ before taking the convolution.

## 2.2 Comparing model and data

With that, we are finally ready to compare our model to observational spectroscopic data and run an optimisation algorithm. To quantify the agreement between model and observational spectroscopic data and fit for the abundances of CO2 and CH4, we use the approach described in Brogi and Line (2019). In their framework, they build a likelihood function $L$ starting from Pearson's cross-correlation coefficient via the following equation:

$$\log(L) = -\frac{N}{2} \left[ s_f^2 - 2R(s) + s_g^2 \right] \tag{11}$$

---

[8]https://www.hitran.org/cia/#ref





In Equation (11), $N$ is the total number of data points, $s_f^2$ is the variance of the observed spectrum $f(n)$, $s_g^2$ is the variance
of the model spectrum $g(n)$ and $R(s)$ is the cross-covariance, given by, respectively:

$$s_f^2 = \frac{1}{N} \sum_n f^2(n) \tag{12}$$

$$s_f^2 = \frac{1}{N} \sum_n g^2(n-s) \tag{13}$$

$$R(s) = \frac{1}{N} \sum_n f(n)g(n-s) \tag{14}$$

In the previous equations, $n$ refers to each wavelength value in our spectra and $s$ refers to a wavelength shift. In our case,
there is no shift $s$ between the models and the data as the telluric lines are not subject to any significant Doppler shift with
respect to the observer. Hence, we do not include the shift as a model parameter and instead fix $s = 0$. Equation (11) is then
used to drive a Markov-Chain Monte Carlo to estimate the best-fit model parameters, with details about chain length and
convergence criteria explained in Section 4.

## 3 Data

Many different data sets were employed in this work, either used in the modelling process or as a comparison for our results.
All of these are described in more detail in the following sections.

### 3.1 CARMENES data

The main goal of our new synthetic transmission algorithm is to fit telluric lines present in stellar spectra. Thus, we required
spectroscopic data taken with ground-based telescopes. Even though archival data would suffice, we decided to carry out new
observations so we could simultaneously measure our own atmospheric profile and ground $CO_2$ abundance with a weather
balloon launch and a $CO_2$ sensor, described in Section 3.2.2. The different atmospheric profiles used and how they were
employed are described in Section 3.2. The spectroscopic data comes from the CARMENES spectrograph (Quirrenbach et al.,
2014), mounted in the 3.5m telescope at the Calar Alto Observatory, located in the southern part of Spain (coordinates $37.22°$N
and $2.55°$W). Observations were carried out on the night of April 24th 2023.

CARMENES has two separate échelle spectrographs, one covering the visible wavelength range, from $0.55 - 1.05$ $\mu$m, and
the other covering the near infrared range, from $0.95 - 1.7$ $\mu$m. We are interested in the latter one, as it is in the near infrared
that we encounter prominent $CO_2$ and $CH_4$ telluric lines. The resolving power for CARMENES in the NIR is reported to
be $R = 80400$ (Quirrenbach et al., 2018) and is obtained from the average of measurements of unresolved lines of a hollow
cathode lamp (Quirrenbach et al., 2016).



| Order | Wavelength range ($\mu$m) | Molecular lines |
|---|---|---|
| 12 | $1.16 - 1.19$ | H2O |
| 16 | $1.26 - 1.28$ | O2 |
| 25 | $1.55 - 1.58$ | CO2 |
| 26 | $1.59 - 1.62$ | CO2 |
| 27 | $1.64 - 1.66$ | CH4 |
| 28 | $1.68 - 1.71$ | CH4, H2O |

**Table 1.** Summary of CARMENES orders used in the analysis, along with the wavelength range they cover and the prominent molecular lines found in this range for the molecules relevant to this study.

As a target, we used a telluric standard star. Based on the visibility conditions at the time interval granted to us, the selected target was HR 5676, an A type star with $V = 5.272$ (Høg et al., 2000). In total, 66 observations were carried out throughout the night, between UT 20h57 and UT 03h57, with 68 seconds exposures for the NIR observations.

     CARMENES data products come in pairs, one corresponding to the data obtained by the science fiber A and the other corresponding to the data obtained by the calibration fiber B, which in most cases is pointed to the sky. Here, we only use the

data coming from fiber A. The science data is divided in orders, each covering a portion of the whole wavelength range. There are 28 orders in total, and for this analysis we only used 6 of them, namely the ones that contained prominent absorption lines associated with the molecules relevant to our study. Table 1 gives a summary of which orders were used, their wavelength range and which molecules contain prominent lines in this range.

### 3.1.1   Normalisation process

The model spectra described in Section 2 is computed normalised to 1, where 1 corresponds to total transmission and 0 corresponds to zero transmission. For the observational spectra, however, that is not the case, as the flux level the spectra come is affected by a number of sources, such as instrumental systematics and physical effects like stellar variability. Therefore, both our model spectra and the observational spectra need to be normalised before they can be compared to each other.

     The normalisation process involves creating a "normalisation mask" that is used to determine what is a spectral line and what

is part of the continuum, and then running a median filter only on the continuum points. The normalisation mask is created from a dummy model spectra containing arbitrary values for the molecular abundances of CO2 and H2O, listed in Table A1. A first median filter is run through the whole spectra (including the lines) to bring everything down to zero. Then, we calculate the median absolute deviation (MAD) of the spectra, and everything that is below a certain multiple of the MAD is considered a line.

This mask is then used to normalise the observational spectra and the model spectra in all steps of the MCMC. The normalisation is done by running a second median filter only on the masked dataset, which hopefully should include only the continuum points, and then linearly interpolating it to fill in the gaps left by the lines. Finally, the normalised spectra is obtained by simply dividing the original spectra by the interpolated median filter.





In this process, there are certain decisions for parameters that can affect the results of the analysis. They are:

1. the rule that defines what is a line and what is the continuum;

2. the abundance values used to generate the dummy model spectra used for the normalisation mask;

3. the window size for the first median filter;

4. the window size for the second median filter.

For CARMENES, the spectra is divided in orders, so we have a different window size value for each order. A summary of the values used for each of these rules is shown in Table A1.

All of these parameters, however, did not prove to significantly alter the results, although a statistical analysis would need to be carried out to more robustly assess their actual influence on the model. This may be tackled in future work.

### 3.1.2 Emission lines

With the normalisation done, the model spectra is ready to be compared to the observational spectra, but the observational spectra still requires some further tweaks. Our model does not include emission lines, so they must be removed from the observational spectra to avoid any issues. This is done by using telluric emission spectra from the ESO Sky Model Calculator (Noll et al., 2012; Jones et al., 2013). Details of this data set are given in Section 3.3. The emission spectra is regridded to the same wavelength distribution as the observational spectra, and then a mask is created to identify the positions of the emission lines. This is done by defining a flux level above which everything is considered to be an emission line. The points considered to be emission lines are then masked out of the spectra. If we are too conservative, we might not remove certain emission lines that could cause trouble in our analysis, but on the other hand, if we are too lenient, too much of the spectra might end up being removed, along with some absorption lines, in observations where emission lines might not even be present. There is one minimum flux level defined for each CARMENES order used, which are listed in Table A1.

### 3.1.3 Deep and saturated absorption lines

Throughout the analysis, another aspect that has proven to be troublesome is absorption lines that are too deep. When absorption gets higher, the signal gets progressively lower, such that the signal from deep lines may be comparable to the noise in the data, plus saturated lines are problematic because information is lost. Therefore, we decided to not include points below a certain flux level, which was chosen to be 0.2, in the calculation of the log likelihood function. This rule is also included in Table A1.

### 3.2 Atmospheric data

Atmospheric conditions can vary drastically depending on the time and place. These will affect how light interacts with molecules in the air, thus influencing the resulting telluric spectra. To properly model the spectral lines, we require temperature, pressure and molecular abundance values as a function of height, which collectively we refer to as the "atmospheric profile".





### 3.2.1 Literature atmospheric profiles

The Michelson Interferometer for Passive Atmospheric Sounding (MIPAS; Fischer et al., 2008)[9] is a mid-infrared emission spectrometer onboard the ENVISAT (Louet and Bruzzi, 1999) satellite. MIPAS has atmospheric profiles that provide all of the aforementioned parameters, but they are only computed for general locations such as mid-latitude (day and night), polar winter/summer and equatorial day-time rather than for specific locations, plus there is no time information on the profiles.

On the other hand, the Global Data Assimilation System (Kanamitsu, 1989; Derber et al., 1991, GDAS;)[10], provided by
the US National Oceanic and Atmospheric Administration (NOAA), has profiles for specific locations around the globe, the aforementioned Calar Alto Observatory being one of them. Additionally, their profiles are computed every 3h, so we can get conditions very close to the time of observation. However, the GDAS profiles only contain information on the molecular abundance of water (i.e., the humidity), so the MIPAS profile is still required to obtain information on the other molecules.

The GDAS profiles span an irregular grid from $\sim 0.1 - 26$ km, whereas the most recent MIPAS profiles go from $0 - 200$
km in constant steps of 1 km. Both the MIPAS and GDAS profiles come from actual measurements of the atmosphere. MIPAS used to observe different atmospheric levels at a line of sight that penetrated down to a minimum altitude, and the profiles were obtained from a forward retrieval model (Fischer et al., 2008), whereas the GDAS profiles combine different types of observations such as surface observations, balloon data, aircraft reports and satellite observations (Rodell et al., 2004).

### 3.2.2 Weather balloon launch

To complement these data sets, we decided to measure our own atmospheric profile using a weather balloon. The balloon launch was carried out on the same night as the spectroscopic observations described in Section 3.1. All of the materials necessary for the launch were purchased from Stratoflights[11], who provide a complete kit that contains all of the essentials to carry out such an experiment. The balloon selected could withstand a payload of up to 2000 g and had a burst altitude of around 38 km. The electronics responsible for carrying out the measurements were the DataloggerSTRATO4. This device contains two
sets of sensors, one of which remained inside the balloon probe (a cube polystyrene box) and the other which was outside of the probe. The internal sensor was responsible for measuring the location (latitude, longitude and altitude), ground speed and temperature, as well as keeping track of time, whereas the external sensor measured pressure, humidity and temperature. Measurements were taken with a 2 second cadence.

Alongside the datalogger, the balloon probe also contained two GPS trackers, which were essential for retrieving the balloon
once it landed. Initially, we planned to launch the balloon as close to the observatory as possible, such that the data collected was as local as it could be. However, our flight path simulations[12] showed that by doing so, the likelihood of the balloon landing in the ocean was very high, which would have made it considerably harder, if not nearly impossible, to retrieve the balloon with the measurements intact. Therefore, we had to compromise and move our launch site. Taking all of the logistics

---

[9]eodg.atm.ox.ac.uk/RFM/atm/

[10]ftp.eso.org/pub/dfs/pipelines/skytools/molecfit/gdas/

[11]www.stratoflights.com/en/

[12]www.stratoflights.com/en/tutorial/weather-balloon-tools/predicting-the-flight-path/



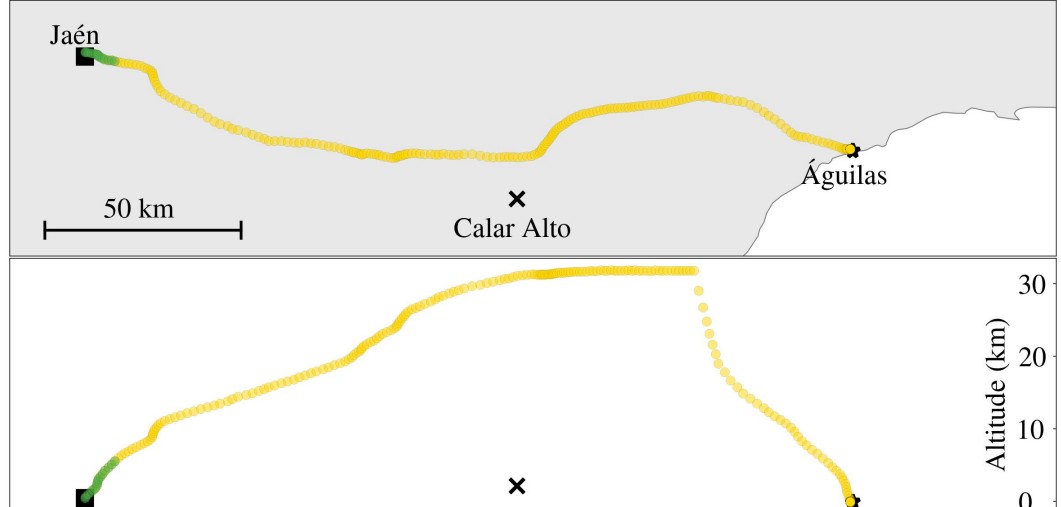

**Figure 1.** Top: flight path of our weather balloon. Locations of the Calar Alto Observatory, the city of Jaén (launch site) and the municipality of Águilas (crash site) are indicated by the black cross, the black square and the black star, respectively. Map scale is shown on the bottom left corner. Bottom: altitude variation throughout the flight. Symbols indicate the altitudes of the aforementioned sites. The green circles correspond to the points prior to the datalogger malfunction, while the yellow circles are the points after that. For visualisation purposes, the data set was plotted with a 1m40s cadence instead of 2s.

and safety measurements into account, the selected launch site was the city of Jaén, around 125 km away from the Calar Alto
Observatory. Even with our extra precautions, and despite being more than 100 km from the nearest shore, the balloon almost landed in the ocean, in the municipality of Águilas, as can be seen from Figure 1, which shows the flight path of our weather balloon.

Due to an unknown electronics malfunction that happened during the flight, the external sensor stopped recording data around 30 minutes after launch, when it had risen to a height of around 6 km. The trajectory of the weather balloon prior to the
electronics malfunction is represented by the green circles in Figure 1.

### 3.2.3 Combining the atmospheric profiles

In order to get the most accurate atmospheric profile possible, we employ all the profiles at our disposal. That includes site data, balloon data, GDAS data and MIPAS data. The site data is collected by the observatory's weather station and included in the file header of each observed spectrum. The combined atmospheric profile consists of site data plus whatever profiles
we choose to attach on top of it. To ensure a smooth transition between the site data and the subsequent profile, an approach similar to that described in Noll et al. (2013) was applied, which adjusts the contribution of the site data by a scaling factor for heights $h$ below a critical height $h_{\mathrm{crit}}$. The corrected profile value is calculated as:



$$\nu_{\text{profile,corr}}(h) = c(h) \times \nu_{\text{profile}} \quad , \tag{15}$$

where the correction factor $c(h)$ is given by:

$$c(h) = \begin{cases} c(h_{\text{site}}), & h < h_{\text{site}} \\ (1-b)\frac{h}{h_{\text{crit}}} + b & h_{\text{site}} < h < h_{\text{crit}} \\ 1, & h > h_{\text{crit}} \end{cases} \quad , \tag{16}$$

where

$$b = \frac{c(h_{\text{site}})h_{\text{crit}} - h_{\text{site}}}{h_{\text{crit}} - h_{\text{site}}} \tag{17}$$

and

$$c(h_{\text{site}}) = \frac{\nu_{\text{site}}}{\nu_{\text{GDAS}}(h_{\text{site}})} \tag{18}$$

For Cerro Paranal, an inspection of the GDAS data showed that there is a gradual reversal of wind direction from sea level up to around 5 km, beyond which it remains constant, as reported in the documentation for the Cerro Paranal Advanced Sky Model (Noll et al., 2013). Due to this feature in the atmospheric data, it is argued in the documentation that it is safe to assume that above this critical altitude the air properties are no longer significantly correlated with the local site measurements and are described by the GDAS data only. The altitude of 5 km is roughly twice the height of the Cerro Paranal Observatory, therefore to determine the critical height for other observatories we decided to use twice the observatory height.

Here, we will refer to using the corrected profile values to merge site and profile data as the "complex" site merging approach. In contrast, simply plugging the profile values on top of the site values is referred to as the "simple" site merging approach.

Since the MIPAS profile is computed for a generic latitude and has no time information, we deem it as "less representative" of the conditions at the time of observation. Between the GDAS and the balloon profiles, either could be argued as the best option, so we explored both options before settling with our final combined atmospheric profile. This process is described in Section 4.1. To keep computational time manageable when using the balloon profile, the original 2-second cadence of the datalogger was linearly interpolated to a new grid of 50 height points. This new grid goes from the site height up to the highest point in the top profile. Amongst the available profiles, the one that reaches highest in the atmosphere is the MIPAS profile, which goes up to 200 km. However, beyond a certain point the pressure is so low that it is below the limits of our computed cross-sections. This happens at a height of around 80 km, so we clip the MIPAS profile and keep it only up to there. The points in the new grid are evenly spaced in $\log(h)$, resulting in a smaller spacing for the points closer to the ground, which have a larger influence on the model because pressure and abundances are larger there. The bottom profiles (i.e., the balloon and GDAS profiles) are only interpolated up to their maximum height values to avoid having to extrapolate. In the transition points between profiles, a weighted mean is applied that quantifies the contribution of each profile in order to smoothly go from one to the other. For the Cerro Paranal Advanced Sky Model, they use 80%, 60%, 40% and 20% as the contributions of



the GDAS profile in the transition points between the GDAS and MIPAS profiles (the contribution decreases as the altitude increases). Aiming for a smoother transition, we chose to use the last nine points as transition points, with the bottom profile contribution going from 90%-10% in 10% decrements. After that, the top profile values are used in their entirety. Whenever we use weights to transition from one profile to the other, we call that "smooth profile transition". If no weights are used, then

we call it "discrete profile transition".

This approach was used when merging the balloon profile with the GDAS profile as well as when merging the GDAS profile with the MIPAS profile. The MIPAS profile used is the equatorial one, and as already mentioned, it is the same no matter the observation date. The GDAS profile, however, is specific for Calar Alto and taken every 3 hours. The selected GDAS profile is therefore obtained by the linear interpolation of the two profiles closest in time to the observation.

The abundances for every molecule except water were linearly interpolated from the MIPAS profiles, using the height grid as the scale. Since the other profiles had their own humidity (i.e. water abundance) values, we used them instead. Figure 2 shows the resulting atmospheric profiles for all of the 66 CARMENES observations when we employ all three available profiles and combine them using the complex site merging and the smooth profile transition.

### 3.3 ESO Sky Model data

Another necessary step in our modelling process is to remove emission lines from the observational spectra, since our model does not include them. In order to do that, we rely on data from the ESO Sky Model Calculator (SKYCALC)[13], a web application based on the Cerro Paranal Advanced Sky Model (Noll et al., 2012; Jones et al., 2013, ASM;) that allows one to generate both telluric transmission and emission spectra.

Among the parameters that can be changed in SKYCALC, we have observatory height, airmass, season, period of night,
precipitable water vapour (PWV), wavelength range and binning, and whether to convolve it with a line spread function to broaden the spectra or not. For our current spectroscopic data sample, we have verified that using the same emission spectra template works for all observations. This is because sky emission is insensitive to PWV changes and CARMENES is a stabilised instrument, resulting in a constant line spread function.

The template used had the default values for SKYCALC, which are set to an observatory height of 2640 m (Cerro Paranal),
airmass of 1, season and period of night are entire year and entire night, PWV = 2.5 mm, a logarithmic binning of $\frac{\lambda}{\Delta\lambda} = 20000$ and no broadening. The default wavelength range is $1 - 2 \ \mu$m, but we limited it to $1 - 1.8 \ \mu$m since there was no need to go all the way to $2 \ \mu$m. The details on how the emission spectra were used in the modelling process are described in Section 2.

Transmission spectra from SKYCALC were also employed in this work. We used them as reference to test the capabilities of our model. This process and more information on the transmission spectra generated are described in Section 4.2.

### 3.4 OCO-2, OCO-3 and GOSAT-2 data

To obtain the abundance of a certain molecule at a specific height (for example, at the height of the Calar Alto observatory), we require an abundance profile. Naturally, the choice of profile can potentially affect what we call the ground-level abundance.

---

[13]*https://www.eso.org/sci/software/pipelines/skytools/skymodel*



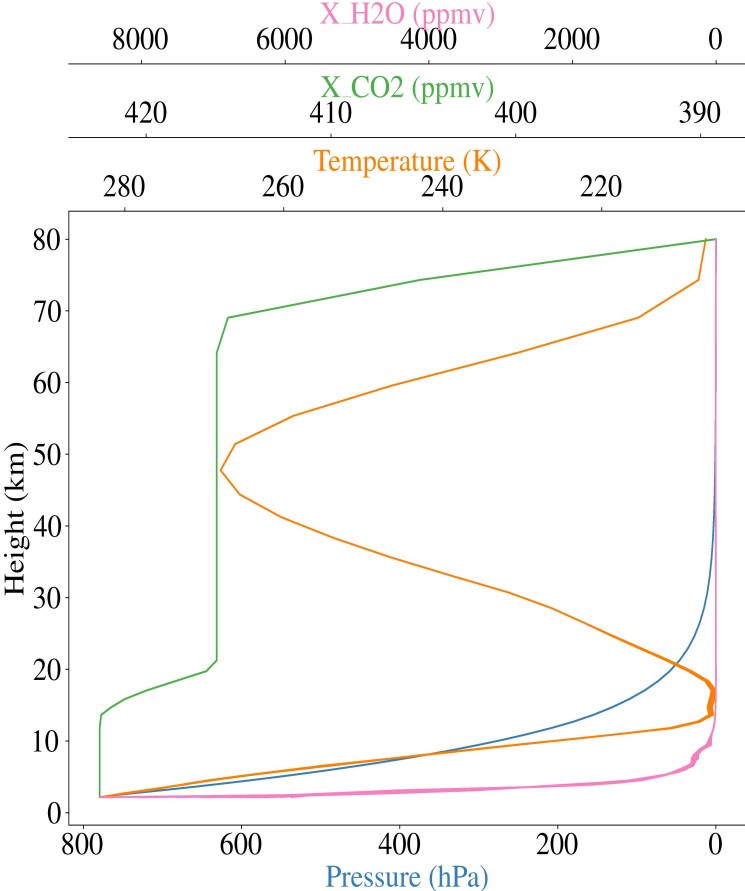

**Figure 2.** Atmospheric profiles for each of the 66 observations taken with CARMENES. Pressure, temperature, CO2 abundance and H2O abundance are shown as a function of height in blue, orange, green and pink, respectively. The profiles are obtained by combining site data, balloon data, GDAS data and MIPAS data using the complex site merging approach and smoothing out the transition between profiles.

Since our method is not the most suited to constrain the abundance profile shape, a better parameter to compare our results to is the total column abundance of CO2 along a certain line of sight. This is measured by satellites such as those from the

Orbiting Carbon Observatory mission, OCO-2 and OCO-3, and GOSAT-2, as well as by the ground-based network TCCON. Here, we employ data from the three satellites to obtain alternative CO2 profiles and we also use their column CO2 retrievals as reference values for our results.

    The column abundance of CO2 can be parameterised by the column-averaged CO2 dry air mole fraction (DMF), denoted $X_{\mathrm{CO2}}$ and defined as the ratio of the column abundances of CO2 to the column abundance of dry air:

$$X_{\mathrm{CO2}} = \frac{\int_0^\infty N_{\mathrm{CO2}}(z)dz}{\int_0^\infty N_{\mathrm{air}}(z)dz} \quad , \tag{19}$$



where $N_{CO2}(z)$ is the altitude dependent number density of CO2, that is, the number of CO2 molecules per cubic meter, and $N_{air}(z)$ is the same quantity for dry air. Equation (19) can be rewritten in terms of the number density of oxygen since the dry air mole fraction of O2 in the atmosphere, $DMF_{O2}$, is a known quantity and essentially constant (Crisp, 2015), so we get:

$$X_{CO2} = DMF_{O2} \times \frac{\int_0^\infty N_{CO2}(z)dz}{\int_0^\infty N_{O2}(z)dz} \quad , \tag{20}$$

The DMF of O2 has been reported to be 0.2095 in papers related to GOSAT (Morino et al., 2011) and TCCON (Laughner et al., 2023a), but 0.20935 in documentation related to OCO-2 and OCO-3 (Crisp, 2015; Crisp et al., 2020). Here, we will use $DMF_{O2} = 0.20935$.

The measurement approach for OCO-2 and OCO-3 consists in collecting spectra in eight adjacent spatial footprints at 1/3 second intervals, and the spectra is obtained combining simultaneous measurements from three spectral regions to define one
"sounding", so in total there are 24 soundings per second (Crisp et al., 2020). The three regions are two CO2 bands, one near 1.61 $\mu$m (weak) and the other near 2.06 $\mu$m (strong), plus the O2 A-band at 0.765 $\mu$m (Crisp et al., 2017). The two CO2 bands provide information about the column CO2, with the strong band also providing aerosol, water and temperature information, while the O2 band provides information about the cloud/aerosol coverage and about the surface pressure (Basilio et al., 2014; Crisp, 2015; Crisp et al., 2020).

However, in order to turn these spectroscopic measurements of the three bands into a calculated value for the column-averaged dry air mole fraction of CO2, both OCO-2 and OCO-3 require a CO2 profile. This is a retrieved parameter from their algorithm obtained from scaling an a priori profile that is used as a starting point. This a priori profile is based on measurements from the NOAA Mauna Loa and American Samoa observatories. They assume that the concentration of CO2 at a certain time and place is a function of the CO2 concentration at these sites and the time of transport between them and the profile location,
also including chemical loss in the stratosphere. This approach is similar to the one employed by the GGG2020 software developed by the TCCON (Laughner et al., 2023a, b).

As for GOSAT-2, it gathers data in three bands in the short wavelength infrared (SWIR), whose ranges are $0.75 - 0.77$ $\mu$m, $1.56 - 1.69$ $\mu$m and $1.92 - 2.33$ $\mu$m, and in two bands in the thermal infrared (TIR), whose ranges are $5.5 - 8.4$ $\mu$m and $8.4 - 14.3$ $\mu$m. The first SWIR band is used to retrieve the surface pressure, the second SWIR band is used to retrieve $X_{CO2}$
(from the 1.6 $\mu$m CO2 band) and $X_{CH4}$ (from the 1.67 $\mu$m CH4 band), and the third SWIR band is used to retrieve $X_{CO2}$ and $X_{H2O}$ (from the 2.08 $\mu$m CO2 band), and $X_{CO}$, $X_{CH4}$ and $X_{H2O}$ (from the 2.3 $\mu$m CO band). Observations are carried out approximately every 4.65 seconds, and the observed area is equivalent to a circle with a diameter of around 9.7 km at the nadir point of the satellite (Imasu et al., 2023).

Similarly to OCO-2 and OCO-3, GOSAT-2 starts with a priori profiles of the relevant molecules, which in this case are CO2,
CH4, H2O and CO. The first two come from NICAM-TM (Niwa et al., 2011), H2O comes from JRA-55 (Kobayashi et al., 2015; Harada et al., 2016), and CO comes from MOZART (Emmons et al., 2010). The retrieved profiles and subsequently the retrieved column-averaged dry air mole fractions are obtained by using the maximum posterior (MAP) analysis method as described in Imasu et al. (2023).



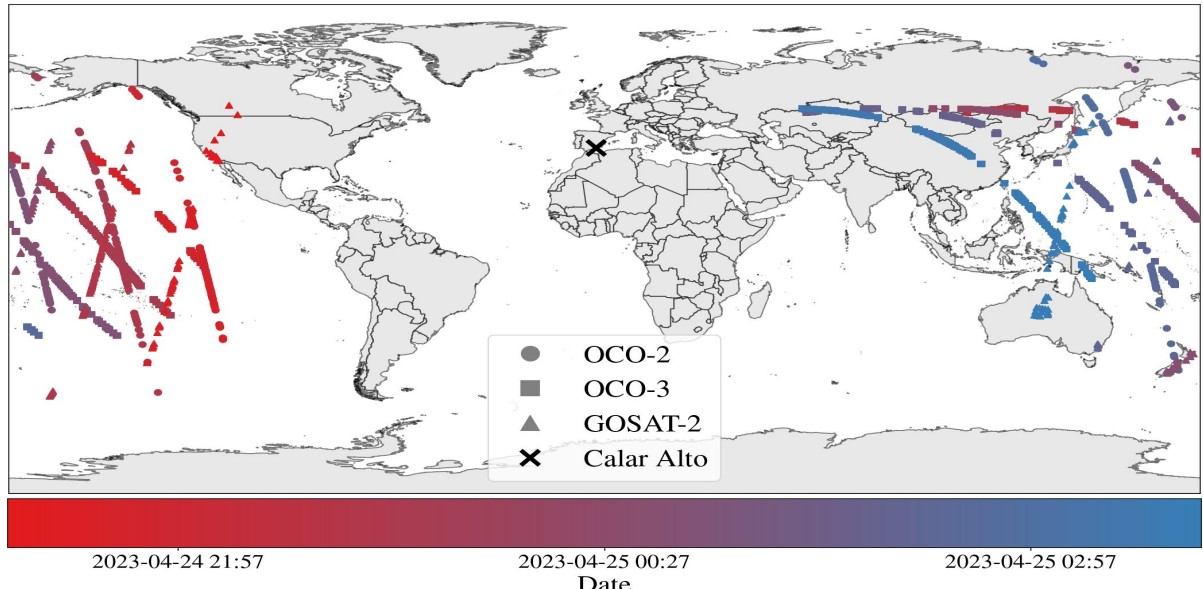

**Figure 3.** Representation of the flight path of OCO-2 (circles), OCO-3 (squares) and GOSAT-2 (triangles) during our CARMENES observations on the night of April 24th 2023, with the location of the Calar Alto observatory marked with a black "X". The data points are colour coded with respect to the time of observation, shown by the colorbar at the bottom.

When comparing our results with those from the aforementioned satellites, there is a compromise to be made, which is
whether to choose their measurements that are closest in time or the ones that are closest in space to ours. The OCO-2 satellite orbits the Earth following the 705 km Afternoon Train (or A-Train) constellation, while OCO-3 is onboard the International Space Station (ISS) and GOSAT-2 is on a Sun-synchronous orbit with an altitude of 613 km, so all three instruments can cover most of the globe. Our observations were taken on the night of April 24th 2023, so we retrieved satellite data from April 23rd UT 00h00 to April 26th UT 23h59. At the exact same time of our astronomical observations, however, none of the satellites
was close to the Calar Alto observatory, passing through the Pacific Ocean and some countries in Asia, as shown in Figure 3. Their closest approach to the observatory in the selected time period occurred on April 26th at UT 13h22min for OCO-2, April 23rd at UT 11h25min for OCO-3 and again April 26th at UT 13h37min for GOSAT-2, as shown in Figure 4. Data from these closest approaches were used to obtain alternative $CO_2$ profiles that were part of our atmospheric profile study described in Section4.1.1.

To compare our calculated $X_{CO_2}$ with the ones from OCO-2 and OCO-3, we gathered all of their available data close to Calar Alto during the period from six months before to six months after the observation date, so between October 2022 and October 2023. The area deemed to be "close to Calar Alto" was arbitrarily selected to be between latitudes 37.0°N and 37.5°N and longitudes 3°W and 2°W. For GOSAT-2, however, there were no measurements taken in this region during the time selected time period, so we had to loosen our criteria and instead chose measurements between latitudes 35.0°N and




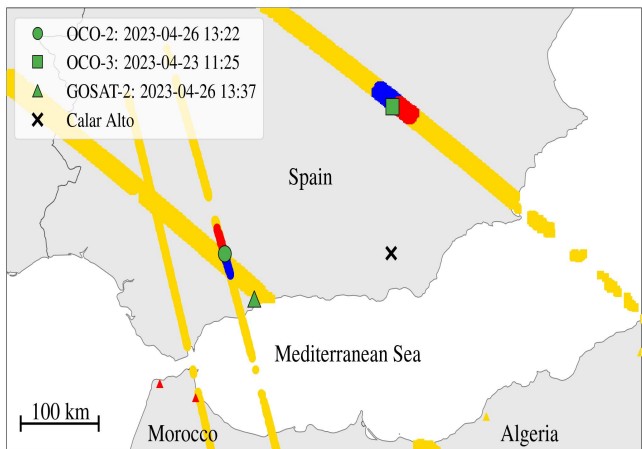

**Figure 4.** Representation of the flight path of OCO-2 (circles), OCO-3 (squares) and GOSAT-2 (triangles) between April 23rd and April 26th, with the location of the Calar Alto observatory marked with a black "X". The red and blue points correspond to the 100 points before and after the closest approaches, respectively, which are marked by the green points. Map scale is shown on the bottom left corner.

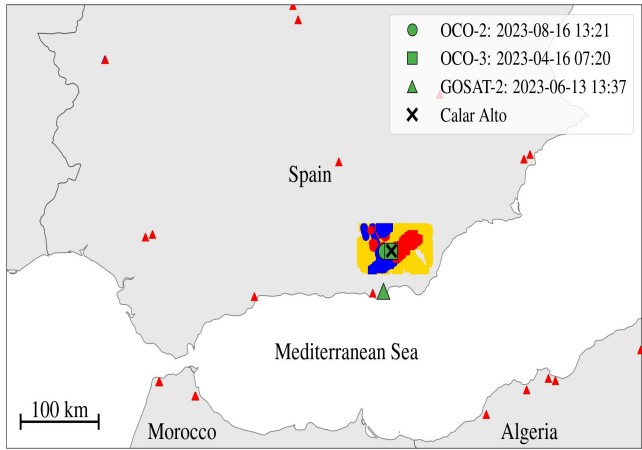

**Figure 5.** Same as Figure 4, but for the satellites' datasets between October 2022 and October 2023 taken close to the Calar Alto Observatory. Map scale is shown on the bottom left corner.

40.0°N and longitudes between 8.0°W and 1.0°E, which is the total area shown in Figure 5. It should be noted that since these satellites' measurements rely on sunlight, there are all daytime measurements.

## 4    Results and Discussion

Our resulting synthetic transmission model contains telluric lines associated with the selected molecules. Based on the overlap between the models for which we had both cross-sections and abundance profiles, the molecules included in the model were



CO2, CH4, H2O, O2 and N2. The first two are the GHGs we are interested in studying and the other three are molecules abundantly present in the Earth's atmosphere. The fitting algorithm developed aimed to find the abundance values from these molecules that resulted in the best fit when compared to observational data. Because N2 does not contain strong spectral features in the wavelength region used here (Smette et al., 2015), its abundance was kept fixed, so the free parameters included are the abundances of CO2, CH4, H2O, O2.

The molecular abundances are included in the modelling process when we calculate the transmission at each atmospheric layer. Changing the molecular abundances in each layer individually would require too much computational time, plus our measurement approach is not suitable to constrain the altitude dependence of molecular abundances. Instead, we parameterise the free variables by their "ground level abundance", which is the abundance given by their respective profiles at the height of the observatory. This ground level abundance is then used to scale the whole abundance profile. The abundance profiles are

given in parts per million (ppm), and by multiplying them by the number density, given by the ideal gas law (Equation 2), we can convert the abundance to molecules/$m^3$. For practical reasons, we deal with the abundances in absolute units throughout the fitting process, but convert it back to relative units to express our results as that is how they are conventionally expressed. The conversion back to ppm is obtained from the definition of the dry air mole fraction, which is "moles of trace gas per moles of dry air" (Laughner et al., 2023a):

$$\text{DMF}_{\text{gas}} = \frac{N_{\text{gas}}}{N_{\text{air}}} \tag{21}$$

In the equation above, $N_{\text{gas}}$ is the number density of the trace gas and $N_{\text{air}}$ is the sum of the number densities of all molecules included in the model, except for water (hence "dry air"). From Equation (21), we can get the DMF of all molecules at each atmospheric layer, and then calculate their column-averaged DMF by using Equation (20).

For water, another useful quantity to express abundance is the "precipitable water vapour" (PWV), which is a measure of the
total amount of water vapour contained in an air column above a certain site and it is usually expressed as the height of liquid water in mm that this water vapour would correspond to. This can be calculated by the following expression (Smette et al., 2015):

$$PWV = \frac{M_{\text{H2O}}}{\rho_{\text{H2O}} R} \int_{z_0}^{\infty} \frac{x_{\text{H2O}}(z) P(z)}{T(z)} dz \quad , \tag{22}$$

where $M_{\text{H2O}} = 0.0182$ kg is the mole mass of water, $\rho_{\text{H2O}} \approx 10^3$ kg/m$^3$ is the density of liquid water, $R = 8.31446 \frac{\text{J}}{\text{molK}}$
is the gas constant and $z_0$ is the height at which we are calculating the PWV. Numerically, this is done by employing the `numpy.trapz` function.

The free parameters' posterior distributions are inferred by running a Markov Chain Monte Carlo (MCMC), which is done with the help of `emcee` (Foreman-Mackey et al., 2013). An MCMC generates a specified number of walkers, where each walker starts with different initial guesses for the free parameters, and runs these walkers for a maximum number of steps or





until a convergence criteria is satisfied. At each step, new values for the free parameters are attempted in order to gradually reach parameter values that would yield a model that best fits the data. To quantify the resemblance of our model to the observational data, we employed the log likelihood function given by Equation (11), which comes from Brogi and Line (2019), as explained in Section 2.2.

In every application of an MCMC shown here, we use the same convergence criteria. One of the available statistical parameters in `emcee` is the autocorrelation time, which basically determines the number of steps needed for the chain to "forget" where it started (because the samples in our chain are not independent) (Foreman-Mackey et al., 2013). An MCMC run would be deemed "converged" if the autocorrelation time of each walker was less than the current step number divided by 100 and if the difference between the previous autocorrelation time and the current one was less than 1%. Tests for convergence were carried out every 100 steps.

In the following sections, we describe the three different analyses to which we subjected our fitting algorithm.

## 4.1 Atmospheric profile study

Before settling for a certain combination of atmospheric profiles to use in our final fit against the CARMENES data, we wanted to investigate how our results were influenced by the choices made in generating the combined profile. In particular, we explored the effects of using and not using the balloon data as well as employing the complex and simple site merging approach in both scenarios. Therefore, we had four different models: simple site merging, no balloon data; complex site merging, no balloon data; simple site merging, balloon data; and complex site merging, balloon data. In every case, subsequent profiles were merged with a smooth transition, as this did not have a significant effect on the results.

For each configuration, an MCMC was run with 10 walkers and 5000 maximum steps. Using a burn-in of half the total amount of steps, the results were calculated as the median of the posterior distribution, with the uncertainty being the standard deviation. The resulting ground level $H_2O$ abundances are shown in Figure 6, along with the humidity measurements from the observatory's weather station, whereas the resulting column-averaged "dry air" mole fractions of $H_2O$ are shown in Figure 7.

Figure 6 tells us that the choice of atmospheric profile has a significant influence in our calculated ground level abundance, where "ground level" corresponds to the height at the observatory in question.

However, Figure 7 shows that the total column-averaged DMF of $H_2O$ is not very sensitive to the choice of atmospheric profile. Therefore, without a reliable abundance profile, it is harder to achieve an accurate abundance measurement for a specific height.

Assuming that the observatory's weather station provides the most reliable measurements for the site conditions, moving forward we will employ the atmospheric profile that better matches the site data, which is the atmospheric profile obtained by employing the simple site merge and not using the balloon data (blue points in Figure 6).

### 4.1.1 Exploring different CO2 profiles

Amongst our available atmospheric profiles, the only one which had abundance profiles for species other than water was the MIPAS profile, so the study described in the previous section had no significant impact on the retrieved CO2 values. Thus, we





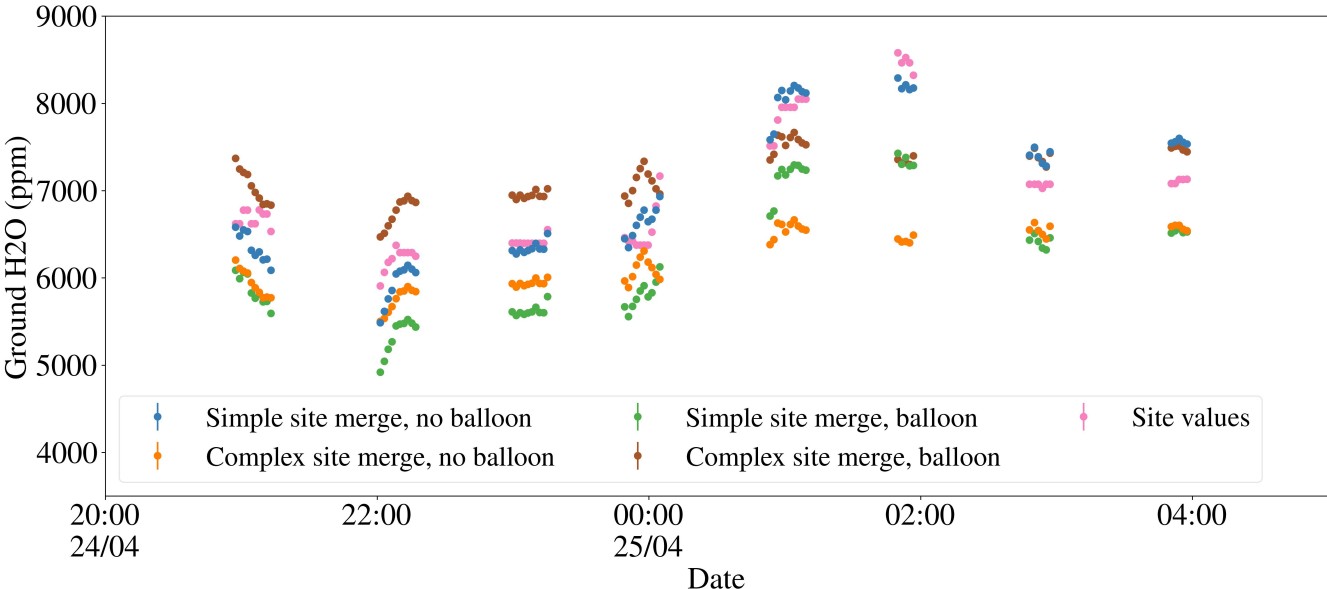

**Figure 6.** Ground level H2O abundance at the night of April 24th 2023 as retrieved by the four different scenarios investigated in our atmospheric profile study: simple site merge, no balloon data (blue); complex site merge, no balloon data (orange); simple site merge, with balloon data (green); and complex site merge, with balloon data (brown). Plotted as well are the measurements from the observatory's weather station (pink).

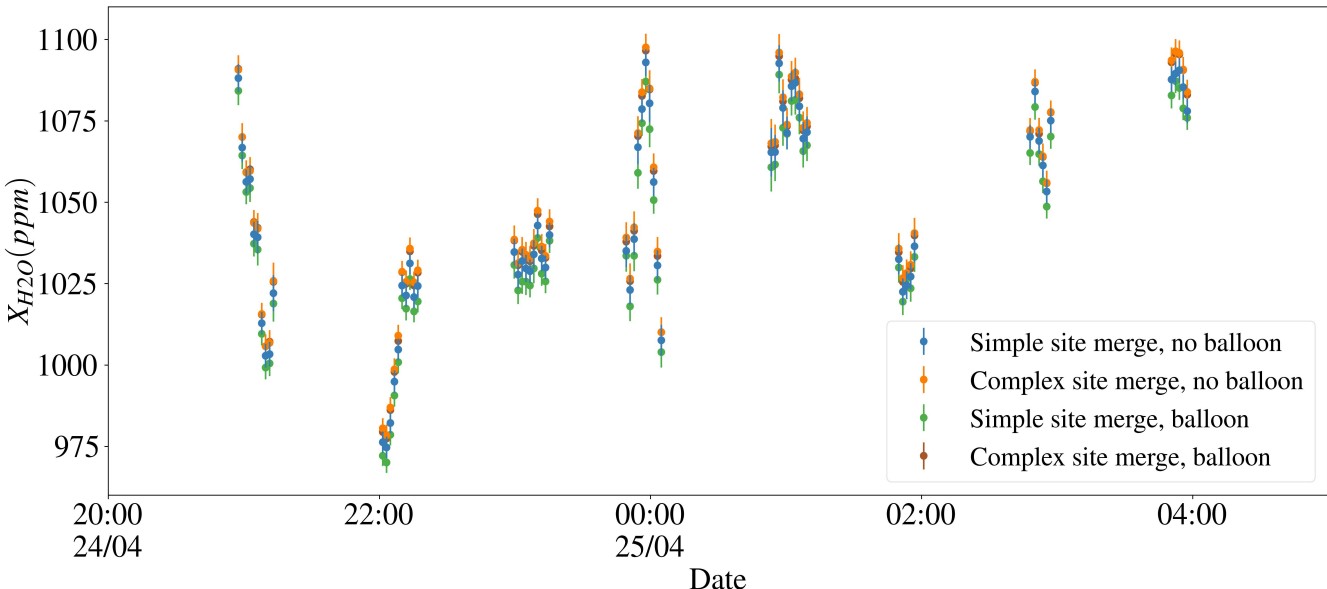

**Figure 7.** Same as Figure 6, but for the column-averaged dry air mole fraction of H2O as given by equation (20).





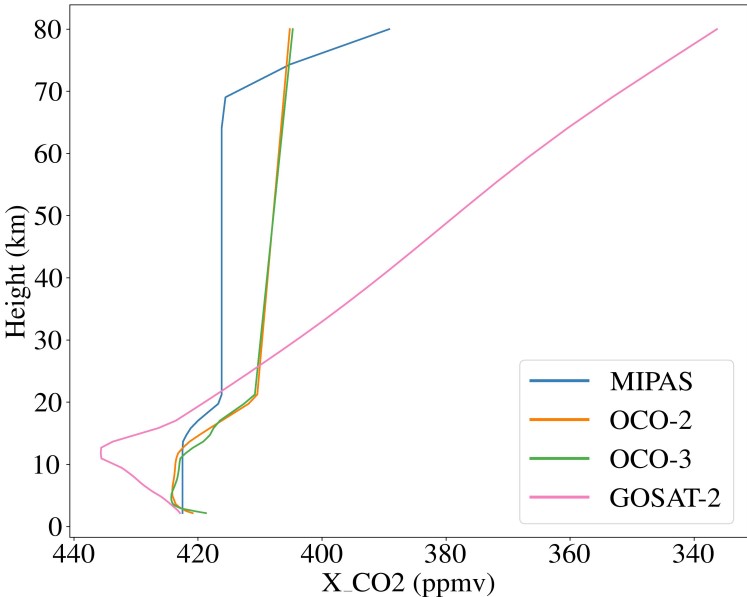

**Figure 8.** Different CO2 profiles employed in the analysis, coming from MIPAS (blue), OCO-2 (orange), OCO-3 (green) and GOSAT-2 (pink). The MIPAS profile was scaled with a factor of 1.13 to roughly match the same ground level value of the other profiles. All of these profiles are scaled during the MCMC, so what is important here is to note how their shapes are slightly different.

decided to verify how changing the CO2 abundance profile directly would affect the results. This was carried out by employing the CO2 abundance profiles from OCO-2, OCO-3 and GOSAT-2.

Four MCMC runs were carried out for this analysis: the first used the MIPAS CO2 profile, the second used an OCO-2 CO2 profile, the third used an OCO-3 CO2 profile and the fourth used a GOSAT-2 profile. The MIPAS profile has no time dependence, but the satellites' profiles vary with time. For OCO-2 and OCO-3, the profiles chosen were the median of the profiles associated with the closest approach shown in Figure 4 and the 100 points before and after (red and blue points in Figure 4). For GOSAT-2, since the precedent and subsequent points were already quite far from the observatory, the profile

employed is simply that associated with its closest approach to Calar Alto between April 23rd and April 26th, which is the green triangle shown in Figure 4. The chosen profiles for each satellite are then linearly interpolated to the same height grid as our combined atmospheric profile before carrying out the MCMC. The resulting CO2 profiles are shown in Figure 8.

Each MCMC had 10 walkers and a maximum number of steps of 5000, with the burn-in being half the total amount of steps. The best-fit values were taken to be the median of the posterior distribution, and from that the ground level abundance and

column-averaged dry air mole fraction of CO2 were calculated. Again, uncertainties were taken as the standard deviation of the posterior distribution. These results are shown in Figures 9 and 10.




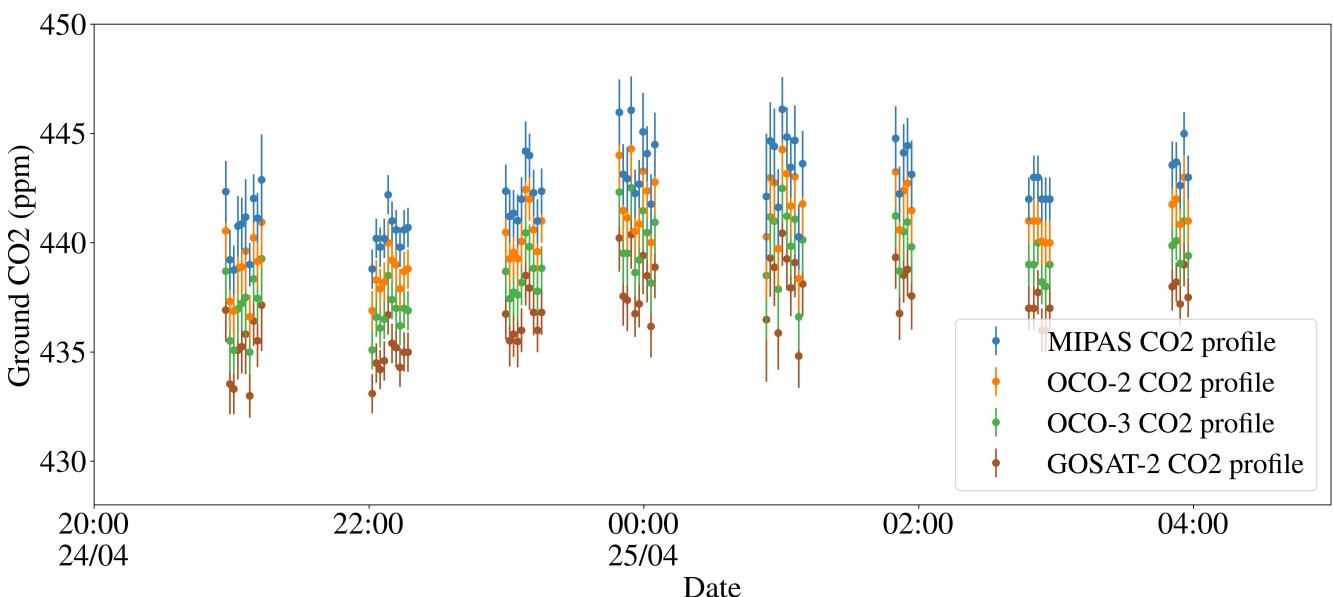

**Figure 9.** Ground level CO2 abundance at the night of April 24th 2023 as retrieved by our analysis when using the MIPAS CO2 profile (blue), the OCO-2 CO2 profile (orange), the OCO-3 CO2 profile (green) and the GOSAT-2 CO2 profile (brown).

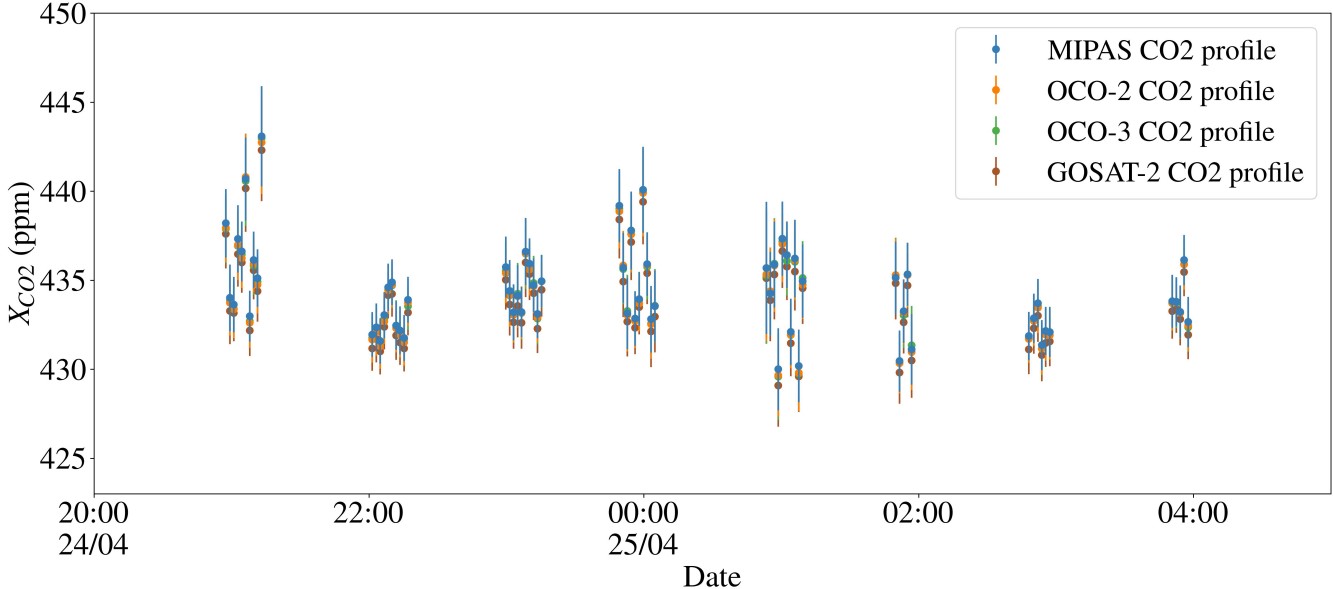

**Figure 10.** Same as Figure 9, but for the column-averaged dry air mole fraction of CO2 as given by equation (20).





Similarly to the analysis in Section 4.1, employing a different profile for CO2 results in slightly different ground level abundances for that same gas, but their retrieved column-averaged DMFs remain mostly unchanged (Figure 10). As expected, the other molecules were not affected by the change in CO2 profile.

From Figure 8, it is not surprising that the ground levels retrieved when using the GOSAT-2 CO2 profile are lower than for the other cases, since this profile has the starkest increase between the abundance at the 2 km mark and the 10 km mark, so with a lower ground level abundance this profile can still yield the same column-averaged dry air mole fraction as the other profiles. This sort of behaviour also explains why the retrievals from using the MIPAS CO2 profile yield the highest ground level abundances. The notable difference between the ground level abundances retrieved by the analyses using the OCO-2 and

OCO-3 CO2 profiles is a bit surprising, given that the profiles are quite similar, but the OCO-3 CO2 profile does appear to have a sharper increase than the OCO-2 CO2 profile at the lowest heights in the profile, so it is understandable that the retrievals for the ground level abundance from the OCO-3 profile are lower than those for the OCO-2 profile.

Since the parameter that is best constrained by our fitting algorithm is the column-averaged dry air mole fraction, and that seemed to be independent of the choice of profile shape, our final results are considered to be those obtained when using the

simplest approach, which is to employ the MIPAS abundance profiles.

## 4.2 Fit to the ESO Sky Model

As a test for our algorithm, we compared it to SKYCALC models. We generated transmission telluric spectra for each of the 13 different available PWV values in SKYCALC: PWV = 0.05, 0.10, 0.25, 0.50, 1.00, 1.50, 2.50, 3.50, 5.00, 7.50, 10.00, 20.00 and 30.00, all in mm. Every other parameter was kept as mentioned in Section 3.3, except the resolution, which we chose to

be $R = 300000$. Starting with the values from the MIPAS profile, we wanted to see if we could reproduce the reported PWV values from SKYCALC. Since our goal was to reproduce the models from SKYCALC, our setup was as close to the one used by ASM as possible and is described in the paragraph below. When changing the PWV values of the ESO models, the abundances of the other molecules should remain unchanged (Noll, private communication).

The atmospheric profile used for this analysis is a combination of the GDAS and MIPAS profiles, plus site data. We selected

the arbitrary date of January 5th 2023 to collect the site data. The MIPAS profile used was again the equatorial one, as this is the recommended one for Cerro Paranal according to the ASM documentation (Noll et al., 2013). The critical height chosen was 5 km, as this is the default for the ASM, and the merging of the GDAS and MIPAS profiles was done with four weights ranging from 80%-20%, again following the documentation (Noll et al., 2013). Because of our arbitrary choice of site values and due to the fact that the ASM interpolates the MIPAS and GDAS profiles to an irregular grid for which the levels are not

stated, our resulting atmospheric profile is slightly different than the one employed to compute the SKYCALC models.

To compare our model to the ESO models, we ran an MCMC with 10 walkers and a maximum of 5000 steps, with a burn-in of half the total amount of steps. The resulting best-fit values were calculated from the median of the posterior distribution, while the uncertainties were obtained from the standard deviation of the posterior distribution, and both are shown in Figure 11 and listed in Table 2.





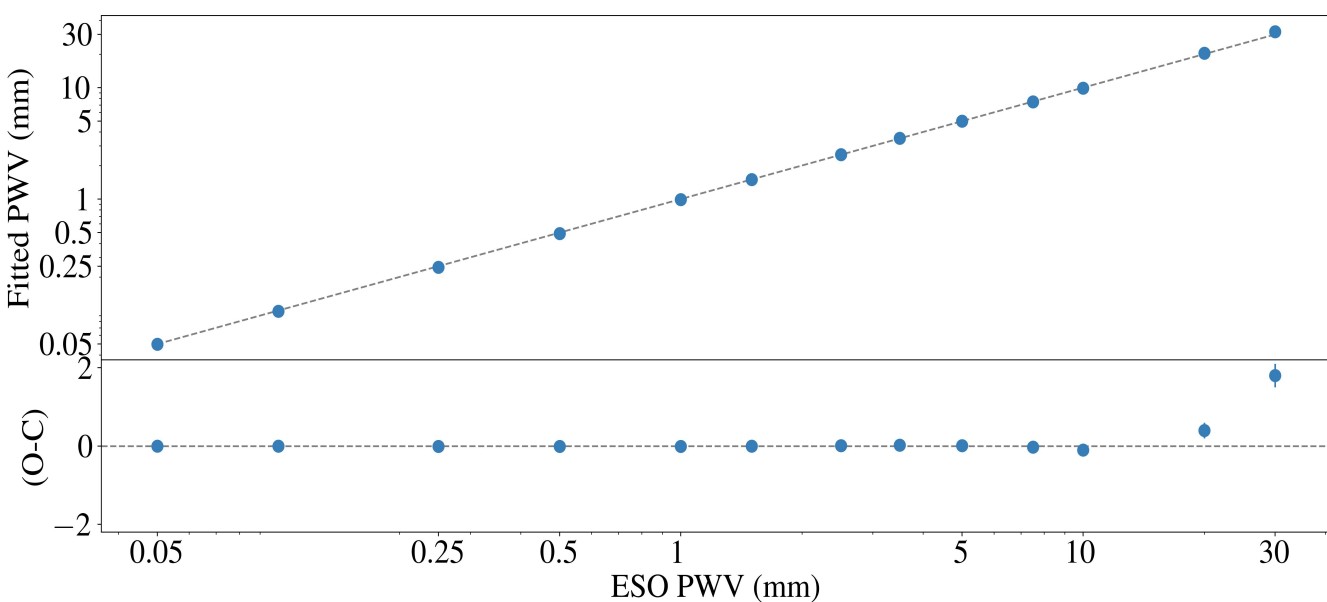

**Figure 11.** Comparison between the ESO reported PWV values and the fitted values from our telluric model algorithm. The dashed grey line corresponds to the identity line.

| ESO PWV (mm) | Fitted PWV (mm) |
|:---:|:---:|
| 0.05 | $0.0499 \pm 0.0001$ |
| 0.10 | $0.0985 \pm 0.0002$ |
| 0.25 | $0.2443 \pm 0.0005$ |
| 0.50 | $0.490 \pm 0.001$ |
| 1.00 | $0.990 \pm 0.004$ |
| 1.50 | $1.496 \pm 0.007$ |
| 2.50 | $2.51 \pm 0.01$ |
| 3.50 | $3.52 \pm 0.02$ |
| 5.00 | $5.01 \pm 0.03$ |
| 7.50 | $7.47 \pm 0.05$ |
| 10.00 | $9.90 \pm 0.06$ |
| 20.00 | $20.4 \pm 0.2$ |
| 30.00 | $31.8 \pm 0.3$ |

**Table 2.** Values and uncertainties associated with Figure 11.





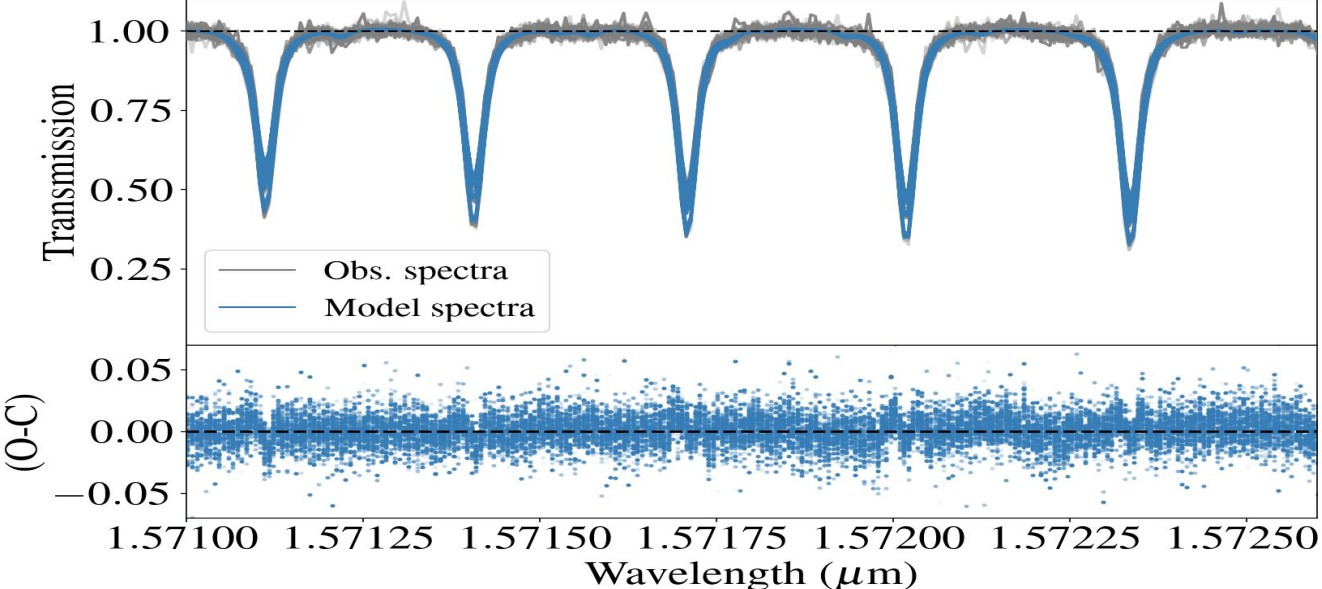

**Figure 12.** Top: best-fit model telluric transmission spectra computed with Astroclimes (blue) and observational spectra from CARMENES (grey) for a portion of the CARMENES order 25, which contains CO2 lines. Bottom: residuals plot. The spectra and residuals shown are for all 66 CARMENES observations.

Our algorithm managed to correctly reproduce the PWV values reported by SKYCALC in all cases except for PWV = 20mm, 30mm, which are quite extreme cases and rarely occur in regions where telescopes are located. The reason for this discrepancy has not been thoroughly investigated, but it might be due to the fact that for such high humidity values most of the water lines are heavily saturated, so it may be hard to distinguish between different abundance values.

### 4.3   Comparison to literature $X_{CO_2}$ values

As shown in Section 4.1, changing the shape of the molecular abundance profiles alters the value we calculate for the abundance of a molecule at a given height, but tends to not affect our measured column-averaged dry air mole fraction, $X_{gas}$, for that molecule. Therefore, that is the parameter we report as the result from our analysis and which we use to compare with literature measurements of the same parameter. The final configuration we opt for employs an atmospheric profile that is a combination of site measurements, GDAS profiles and the equatorial MIPAS profile. The site data is attached to the bottom of the profile

through the "simple site merge" approach and the GDAS and MIPAS profiles are connected with a "smooth profile transition", as described in Section 3.2.3. This corresponds to the blue points shown in Figures 6, 7, 9 and 10, and the CO2 profile shape used in the analysis corresponds to the blue line in Figure 8. An example of how the model spectra look compared to the observational spectra is shown for a region of the spectra containing CO2 lines in Figure 12.



The literature values we use to compare our results to come from the satellites OCO-2, OCO-3 and GOSAT-2, as well as from the ground-based network TCCON. We make use of all the data publicly available from these sources between the period October 1st 2022 UT00h00 and October 31st 2023 UT23h59. For OCO-2 and OCO-3, the data was restricted to that measured inside the latitudes $37°$N and $37.5°$N and the longitudes $3°$W and $2°$W, which corresponds to the small rectangular clump of points in Figure 5. For GOSAT-2, since there were no measurements inside that same rectangle in the specified time period, we loosened our restrictions and included data measured inside the latitudes $35°$N and $40°$N and the longitudes $8°$W and $1°$E, which corresponds to the total area shown in Figure 5. For TCCON, the closest site in their network to the Calar Alto Observatory is Orléans, France (coordinates $47.96°$N and $2.11°$E), so we used data from there (Warneke et al., 2024).

Among the goals of the TCCON network, one of them is "to provide a critically maintained, long-time-scale record to identify temporal drifts and spatial biases in the calibration of space-based sensors" (Wunch et al., 2011). In other words, TCCON is used to calibrate $X_{\mathrm{CO2}}$ measurements from satellites and identify their biases. The TCCON measurements themselves are calibrated by using atmospheric profiles obtained by aircraft flown over TCCON sites (Wunch et al., 2010). For OCO-2 and OCO-3, the bias corrected data is released as part of their data products and it follows the process described in O'Dell et al. (2018). For GOSAT-2, only the bias uncorrected measurements are released, so we had to calculate the bias corrected values ourselves using Equations (1) and (2) from Yoshida et al. (2023) for CO2 and CH4, respectively.

Naturally, we expect our measurements of $X_{\mathrm{CO2}}$ to have biases as well. However, identifying and quantifying these biases is beyond the scope of this paper and will be tackled in future work. For now, we will simply multiply our results by a scaling factor obtained by comparing our values with "truth values". Since the location of the closest TCCON site is quite far from where our measurements were obtained and GOSAT-2 measurements are also very sparse in that area, we will consider our "truth values" to be those from OCO-2 and OCO-3. More specifically, we will only use the measurements taken between a month before and a month after the time of observation to calculate the scaling factor. The bias correction factor $BC$ is then calculated by:

$$BC = \frac{X_{\mathrm{CO2,median,OCO\text{-}2/OCO\text{-}3}}}{X_{\mathrm{CO2,median,Astroclimes}}} \tag{23}$$

This gives us a correction factor of $BC = 0.971 \pm 0.006$, leaving us with a median value of $X_{\mathrm{CO2,median,Astroclimes}} = 421.41 \pm 3.55$ ppm for the distribution of the column-averaged dry air mole fraction of CO2 in the night of April 24th 2023, as measured from our fit to CARMENES data using our newly developed algorithm, Astroclimes. Our bias corrected results compared to those from OCO-2, OCO-3, GOSAT-2 and TCCON are shown in Figure 13.

With our simple bias correction, our results agree well with others in the literature. Additionally, the nightly scatter in our measurements is of the same order of the daily scatter seen on OCO-2 and OCO-3 measurements, though slightly higher than that exhibited by GOSAT-2 and TCCON measurements.

As previously mentioned, GOSAT-2 and TCCON also provide column measurements of CH4, so we also compare our retrieved $X_{\mathrm{CH4}}$ with theirs. For the reasons explained above, we chose not to use GOSAT-2 or TCCON measurements to calculate our bias correction, and since there is no OCO-2 and OCO-3 data for $X_{\mathrm{CH4}}$, here we do not include a bias correction





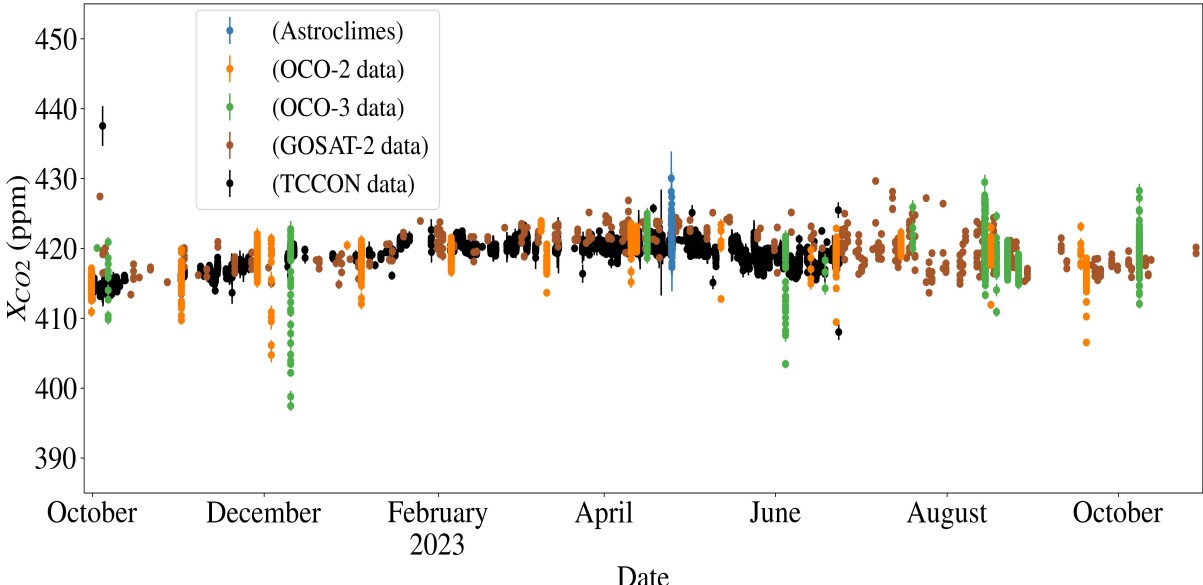

**Figure 13.** Astroclimes retrieved column-averaged dry air mole fraction of CO2 (blue), $X_{CO2}$, along with the same parameter from OCO-2 (orange), OCO-3 (green), GOSAT-2 (brown) and TCCON (black), all bias corrected.

for our retrieved $X_{CH4}$. Our retrieved nightly median for CH4 is $X_{CH4,Astroclimes} = 1.90 \pm 0.02$ ppm. Our retrieved distribution of $X_{CH4}$ along with those from GOSAT-2 and TCCON is shown in Figure 14.

Figure 14 shows that while there is not a significant overall shift between our measurements and those from GOSAT-2 and
TCCON, our scatter is much higher, as well as our uncertainties, the reason for which is unclear.

On average, our $X_{CO2}$ measurements had an uncertainty of 1.6 ppm or 0.4%. For our $X_{CH4}$ measurements, that was 10 ppb or 0.5%. According to Miller et al. (2007), a precision of 1-2 ppm is needed for $X_{CO2}$ measurements "on regional scales to improve our knowledge of carbon cycle phenomena". That is inside our reported precision values, however, even the largest sinks and sources of CO2 rarely produce changes in the background $X_{CO2}$ distribution that exceed 0.25% (Crisp et al., 2017),
so currently our precision falls slightly above that.

On the other hand, Meirink et al. (2006) concluded that $X_{CH4}$ measurements with a precision of 1 to 2% can contribute considerably to reduce the uncertainty in the strength of methane sources, and systematic errors bellow 1% have a dramatic impact on the quality of the derived emission fields. The precision of our $X_{CH4}$ measurements is well within those limits.

### 4.4   Error analysis

$X_{CO2}$ measurements are standardised by the World Meteorological Organization Global Atmosphere Watch (WMO/GAW) CO2 calibration scale (Hall et al., 2021). TCCON measurements are calibrated to this scale via aircraft measurements over TCCON sites and are thus used to validate space-based measurements (Wunch et al., 2010). For OCO-2/OCO-3, this is one



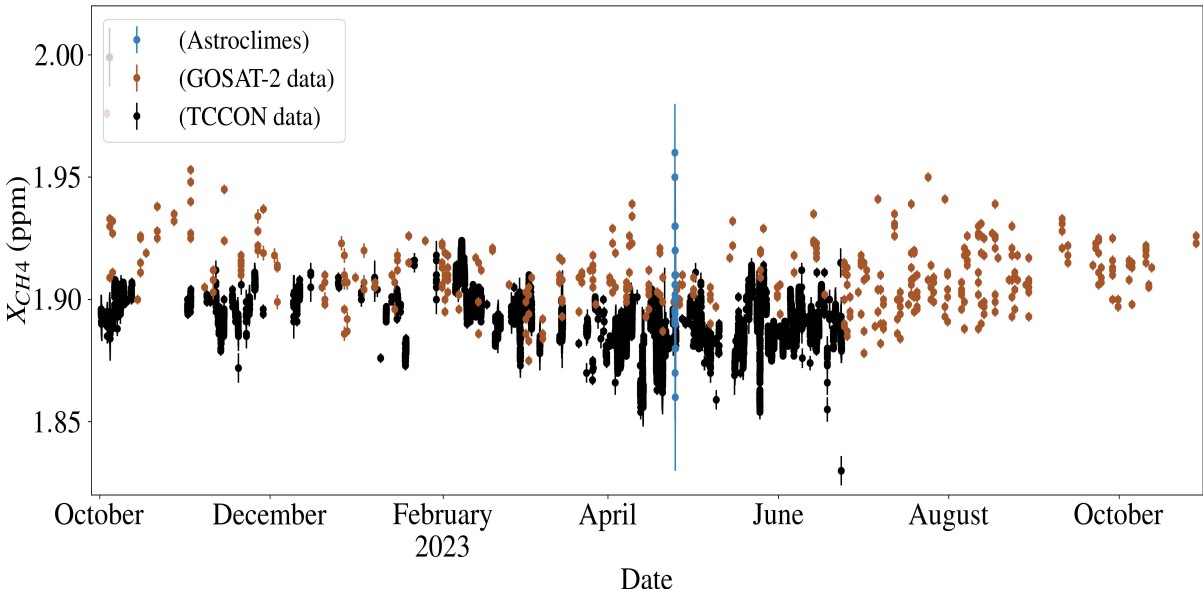

**Figure 14.** Astroclimes retrieved column-averaged dry air mole fraction of CH4 (blue), $X_{CH4}$, along with the same parameter GOSAT-2 (brown) and TCCON (black). Our retrieved values are not bias corrected, whereas the GOSAT-2 and TCCON values are.

of the necessary bias corrections that needs to be carried out to minimise errors, along with corrections for parametric and footprint-level (that is, difference in the measurements from each sensor) (O'Dell et al., 2018). GOSAT-2 measurements also
require correction from systematic biases, an example of which is presented in Yoshida et al. (2023).

Identifying and correcting these systematic biases is a lengthy and ongoing process (O'Dell et al., 2018). We are aware that our retrieved $X_{CO2}$ and $X_{CH4}$ values most likely contain biases, as we have not propagated the uncertainties based on errors on our molecular cross-sections or atmospheric profiles, for example. A follow-up investigation is planned to properly mitigate their effects so that our method can stand on the same ground as those already well established in the literature.

As such, the uncertainties reported in this work do not include systematic biases and are merely precision errors based on the 68% confidence intervals of the MCMC analysis posterior distribution.

## 5  Conclusions

We present a new method to measure the column-averaged DMF of greenhouse gases in the Earth's atmosphere using our newly developed synthetic transmission modelling and fitting algorithm named Astroclimes. While most current ground and
space-based networks only provide column measurements of CO2 and CH4 during the day, as they rely on sunlight, with Astroclimes we can carry out nighttime measurements using telluric standard stars.

Any type of ground-based spectroscopic astronomical observation in the near-infrared can in principle be used to retrieve the column-averaged dry air mole fraction of CO2 and CH4. Since observations of telluric standard stars are a common byproduct



of astronomical spectroscopic observations, we have plenty of archival data in various parts of the world that we can employ.
As such, besides providing column measurements of these gases at night, we can also provide measurements in places that may not be covered by the existing networks. A historical analysis of archival data from several different instruments will be carried out in future work. Using data from before 2021 would also allow us to compare our retrievals with those from the Copernicus Atmosphere Monitoring Service (CAMS) global greenhouse gas reanalysis dataset (Inness et al., 2019; Agustí-Panareda et al., 2023), which should provide a better reference dataset for our night time measurements.

We also obtained new observations using the CARMENES spectrograph in the Calar Alto Observatory in Spain, the reason being because we wanted to carry out a weather balloon launch simultaneously to the astronomical observations, so we could measure our own local atmospheric profile. Unfortunately, due to the logistics of the launch and an electronics malfunction, the weather balloon data was not used in our final analysis. Instead, our atmospheric profile consisted of a combination of site data, GDAS profiles and the equatorial MIPAS profile.

An MCMC analysis was run on our new CARMENES data set to retrieve the column-averaged dry air mole fractions of $CO_2$ and $CH_4$, and a simple bias correction was applied to the $CO_2$ measurements to align them with measurements from the OCO-2 and OCO-3 satellites. With that, our retrieved values show good agreement with the literature, although exhibiting a slightly higher uncertainty (0.4% for $CO_2$ and 0.5% for $CH_4$) than measurements from GOSAT-2 and TCCON. It should be noted, however, that these uncertainties are precision errors from our MCMC analysis and do not include systematic biases. A
more robust and complete bias correction calculation for $CO_2$ and $CH_4$ will be tackled in future work.

**Appendix A: Rules included in the normalisation and data handling process**

*Acknowledgements.* This research was funded in part by the UKRI (Grants ST/X001121/1, EP/X027562/1, MR/S035214/1). The TCCON data were obtained from the TCCON Data Archive hosted by CaltechDATA at https://tccondata.org. We would like to acknowledge the CARMENES staff for carrying out the astronomical observations and showing flexibility and excitement towards our weather balloon ex-
periment, as well as the people from the "Comunidad de Regantes de Águilas" for retrieving our weather balloon and getting in touch with us.





**Table A1.** Summary of the parameters used for the rules applied in the normalisation and data handling process.

| Rule number | What is the rule | Parameters | Values |
|---|---|---|---|
| 1 | Defines line and continuum points | Limiting value for continuum points | $> -1 \times$ MAD |
| 2 | Molecular abundances for dummy model | Ground level $CO_2$ and $H_2O$ abundances | $8 \times 10^{21}$ molecules$/m^3$, $1.5 \times 10^{23}$ molecules$/m^3$ |
| 3 | 1st median filter window sizes | Fraction of total number of points | [0.10, 0.15, 0.10, 0.10, 0.10, 0.10] |
| 4 | 2nd median filter window sizes | Fraction of total number of points | [0.03, 0.01, 0.01, 0.01, 0.01, 0.01] |
| 5 | Defines emission line positions | Minimum emission line height | [500, 100000, 40000, 4000, 2000, 3000]* |
| 6 | Defines minimum transmission included in fit | Limiting value for absorption lines | $> 0.2$ |

* These values were determined based on an emission spectra computed for an observatory height of 2640m (Cerro Paranal), airmass of 1, season and period of night are entire year and entire night, PWV = 2.5mm, a resolution of 20000 and wavelength range from $1.0\mu m - 1.8\mu m$.



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
