# Peer review of "Astroclimes - measuring the abundance of CO2 and CH4 in the Earth's atmosphere using astronomical observations"

_EGUsphere, 2024_

## Author Comment (AC2)

**Astroclimes - measuring the abundance of $CO_2$ and $CH_4$ in the Earth's atmosphere using astronomical observations**
**Response to comments during public peer review**

**Response to Reviewer #1's comments:**

- In the previous short review reply required for starting up the discussion phase, I voted the paper as "good" in all relevant categories, because in my feeling it is important to make the atmospheric community aware of the existence of astronomical observations, which can be useful for the quantification of greenhouse gases (GHG) in the terrestrial atmosphere. As passive methods (specifically solar absorption spectroscopy) are restricted to daytime observations, this might be a useful complementation of existing approaches. I also appreciate the rigorous approach of the authors of even considering a balloon launch carrying instrumentation for in-situ measurement of CO2 mixing ratio and the comparison with TCCON and space borne data.
  On the downside, the authors seem to lack connection with the community doing the operational atmospheric GHG remote sensing observations and therefore seem unaware of the requirements (measurements of column-averaged GHG abundances need to be very accurate to be useful) and the current state of art. This becomes apparent in the manuscript in several places, e.g., MIPAS, a limb sounder, which went out of order more than a decade ago, certainly is not a reasonable source for GHG profiles, as the retrieved column amount critically depends on the tropospheric mixing ratio. The TCCON community meanwhile has established a model-assisted approach for generating quite realistic a-priori GHG profiles.

  - We would like to thank the reviewer for the useful feedback and for making us aware of the possibility of obtaining a priori atmospheric profiles from the TCCON community. We followed the reviewer's suggestion and reached out to the PI of the TCCON Orléans site in the hopes of getting GGG2020 computed profiles for the Calar Alto observatory. Not only were they kind enough to provide us with such profiles for our original observation period, they also assisted us in computing our own GGG2020 profiles, so we were able to generate atmospheric profiles for the whole time range of our extended observational dataset (see response to Reviewer # 2's comments below), which goes from 2016 to 2023. These have been integrated into the analysis and are described in Section 3.3 of the revised manuscript.

- The authors compare the quality of their measurements with the required precision requirements for detecting the large sources and sinks (line 613ff). I would doubt this is a reasonable measure for the required performance. Daytime measurements are much easier to perform and offer a significantly higher precision and accuracy. Dedicated observations using up observation time from huge astronomical telescopes would hardly be competitive with daytime solar spectroscopy, so only the use of calibration measurements performed in the context of astrophysical research purposes remains

as a viable option. What can be learned from nighttime observations? Recently, TCCON has been complemented by COCCON, which uses portable field-deployable spectrometers. These spectrometers can be deployed near interesting sources (as cities, coal mines, landfills, . . . ) for collecting dedicated observations, while astronomical observatories tend to be at remote locations. The variability and trend of background $CO_2$ is easily monitored by the subset of remote TCCON stations.

The remaining interesting gap, which could be filled by astronomical observations, would be the explicit measurement of the complete diurnal $XCO_2$ cycle. This task, however, would require observations of excellent quality and a very high level of consistency between daytime and nighttime observations and the data analysis chains. Most observatories probably reside in regions, which are not too interesting in regard of the diurnal $CO_2$ cycle (remote desert areas), as the interesting signals are created either by biogenic fluxes or by the variable anthropogenic sources. There might be some observatories located in semi-arid high plains, where the study of the biogenic fluxes would be scientifically useful. Some older observatories today located near metropolitan areas might be interesting study places for anthropogenic signals, but probably these sites do not offer the required advanced instrumentation for collecting stellar spectra of sufficient quality.

– Our aim with introducing this new method to measure the abundance of greenhouse gases was always to complement the networks that employ daytime solar spectroscopy, never to compete with them. Observations of the Sun are undeniably easier and it is true that we would not be able to compete with the observational cadence of observatories fully dedicated to observing the Sun. However, the advantage of our observational approach is that our target stars, namely telluric standard stars, can be very bright (in the context of astronomical observations, not compared to the Sun, of course), such that they would only require observations of a couple of minutes, which can be easily introduced to the scheduled of many astronomical telescopes. Telluric standards are also very hot, so their spectra is mostly devoid of spectral features, making it easier to identify and study telluric lines compared to a line-rich spectrum such as the Sun's. Additionally, as was also pointed out by the reviewer, telluric standard stars are frequently observed for calibration purposes, so we have a large archival database at our disposal that would not interfere with the science done by the original PIs. While it is true that most astronomical observatories are located in remote desert areas, we believe they would still provide useful information to existing networks. The Calar Alto Observatory, for example, is relatively close to urban areas. Other telescopes, such as the ones in Mauna Kea and Cerro Paranal, are close to volcanic activity.

- As mentioned before, observations of the diurnal $XCO_2$ cycle would require a very high level of consistency between the nighttime observations and daytime observations. The daytime observations could be realized by collocating a COCCON spectrometer, which offers spectral resolution comparable to the astronomical observations. Furthermore, the same data processing software needs to be used for daytime and nighttime measurements. This could be achieved by extending the data analysis chain as realized by TCCON for handling nighttime observations. For this purpose, spectral

models of telluric standard stars would be implemented in the GGG2020 software to replace the solar source by a telluric standard star source when analyzing nighttime observations. (Concerning the airglow emission lines, the most rigorous strategy would be to perform on/off measurements, then the background spectrum recorded at the same airmass as the stellar observation could be subtracted from the stellar spectrum.) If the authors prefer use of their own software development, the code needs to extended to handle the analysis of daytime observations.**

- – Upon reaching out to TCCON PIs, we were also put in touch with people involved with COCCON, and a potential collaboration is in the works to do precisely what is suggested above (i.e. placing a COCCON spectrometer at an astronomical observatory, but more details cannot be disclosed at this moment). We are also working on tailoring our algorithm to work on TCCON spectra, as well as exploring the possibility of extending the GGG2020 software to work on the spectra of telluric standard stars. However, the format and spectral content of the two datasets are very different to each other and it is not straightforward to implement them in the other algorithm, so we believe this would be substantial work that would be more suited for a separate paper altogether.

- **I suggest a significant revision of the paper, with stronger focus on the description of the astronomical observations, their performance characteristics and availability (this would provide better insight of what can be expected from astronomical observations in general for measuring atmospheric GHG abundances). The use of different codes for nighttime and daytime measurements appears problematic to me, as even a very minor bias would impair the usefulness of adding nighttime measurements. A demonstration of compatibility using collocated daytime-nighttime measurements is required. Finally, particular attention shall be given to ensure consistent data analysis schemes between daytime and nighttime measurement. Implementation of the required extensions for handling nighttime observations into a recognized operational code used for GHG data analysis as GGG2020 would appear more useful to me than the development of own code.**

- – We have followed the reviewer's suggestion and expanded on the description of the astronomical observations at the beginning of Section 3.1 of the paper. While it is true that consistency between the analysis of daytime and nighttime measurements is required, our aim with this paper was to introduce a new method for nighttime measurements, and we will pursue to achieve compatibility with existing daytime measurements. As mentioned in other responses here, we are already working on both tailoring our algorithm to daytime measurements and investigating the capability of expanding the GGG2020 pipeline to work on nighttime measurements. This would require not only testing the compatibility of both algorithms with a very different dataset, but also running the algorithms on these datasets, which is a substantial analysis requiring a dedicated paper. We appreciate the suggestion and are working on this alongside the current work, but do not believe it is critical to demonstrating the potential of the technique, which is the purpose of this paper.

**Response to Reviewer #2's (Matthias Buschmann) comments:**

- **There are mature retrieval algorithms like GFIT, SFIT or PROFFIT used in the TCCON, NDACC and COCCON networks. From an algorithm perspective the authors should first validate their algorithm against any one of these. I'm sure the respective communities will be happy to collaborate and share a set of their spectra to test against. This comparison would evaluate the Astroclimes algorithm performance and using the same or similar prior information and atmospheric parameters has the benefit of reduced overhead in trying to re-invent the wheel (e.g. the used merge algorithm).**
  **In turn, the standard retrievals could be adjusted to work with the presented stellar spectra yielding a robust result derived from recognized standard methods.**

  - We agree that testing Astroclimes against the aforementioned mature retrieval algorithms would provide a robust validation of our algorithm. We have reached out to the PI from the Orléans TCCON site and have obtained some spectra from this site, and the tailoring of our algorithm to work on TCCON spectra is already under investigation. However, after looking into the practicalities of implementing this, we believe that this would require substantial work that should be left for a future paper. To be more specific, the handling of the TCCON spectra before it can be analysed by our algorithm requires some fine tuning in the normalisation process, for example.
    Despite that, we believe this test is not necessary on the present paper to prove the validity of our algorithm. On Section 4.2 of the paper, we test our algorithm against spectra from the ESO Sky Model Calculator. These spectra are calculated using Molecfit (Smette et al. 2015), which is a well-established algorithm used to fit and remove telluric lines from stellar spectra and is widely used in the astronomy community. Our analysis of these spectra yielded very good agreement between our retrieved values and the ESO Sky Model reported values, thus validating our algorithm against a mature retrieval algorithm, albeit not one from the climate community.

- **The comparison/validation presented here has several issues. The required precision to capture the atmospheric variability in CH4 or CO2 requires closer spatial and/or temporal collocation. Thus a comparison to TCCON Orléans is not advisable and great care has to be taken in collocating satellite overpasses. The comparison against available atmospheric reanalysis models of CO2 and CH4 would mitigate some of the issues.**
  **The restriction to measurements from one night is understandable, but if the data exists should be extended to at least a year to test if the relatively well known seasonal cycle of the trace gases can be reproduced or even if any long-term trend is visible.**
  **An additional issue are the reported uncertainties. The variability in the one night of measurements in Fig 13 and 14 respectively suggests a large standard deviation of about maybe 10 ppm (XCO2) and maybe 50 ppb (XCH4) which would mean a relative error of about 2.5% instead of the reported 0.5%.**

– We decided to follow the reviewer's advice and extend our observational sample. We queried the CARMENES database and acquired all of the publicly available data since its first light and filtered through it as described in Section 3.1 of the revised manuscript. Our extended dataset ranges from July 2016 to May 2023.

The decision to only use data from that one night was twofold: we originally intended to combine our weather balloon data with the astronomical observations; and we wanted to first make sure our algorithm worked on a smaller sample before expanding to a larger one. Given the fact that the balloon data was not used in the final analysis, it then made sense to use all of the available data.

The choice of using the TCCON Orléans site and the collocating satellite overpasses was certainly not ideal, but came from a lack of better options, as there are no reanalysis models available for the night of our observations.

With the extended dataset, we used the CAMS global greenhouse gas reanalysis (EGG4) as a benchmark for our results and the comparison is shown in Section 4.3 of the revised manuscript. Even though the EGG4 data is only available until December 2020, it should already be enough to provide a good baseline to compare our results to.

The criticism to our reported uncertainties is valid. Even though we mention that these uncertainties are simply precision errors from the MCMC analysis and do not account for any possible systematic biases, it is true that they do not seem to reflect the exhibited variability in our results. There are two reasons for this: either our uncertainties are underestimated, or the abundances really do vary by that much in such short timescales. We are in no position to argue for the latter, thus we decided to work on our uncertainties now that we have a larger dataset and more suitable and credible benchmark values. This process is described in Section 4.3 of the revised manuscript and we now report a more conservative value for our uncertainties.

- **An additional remark: There have been attempts to perform nighttime measurements from ground-based spectrometers from the TCCON and NDACC communities before. See e.g. https://doi.org/10.1016/S0022-4073(02)00069-9 or https://doi.org/10.5194/amt-10-2397-2017.**
  **I would encourage the authors to connect with any of the TCCON, COCCON or NDACC PIs. The prospect of potentially longer (night) time series of atmospheric spectra of sufficient quality to retrieve trace gases is very much worth pursuing and I encourage re-submission after addressing the mentioned issues.**

  – We would like to thank the reviewer for pointing us to relevant literature and for the words of encouragement. Also, following their recommendation, we reached out to PIs from TCCON and COCCON and new collaborations are underway. More specifically, we have acquired spectra from one TCCON site and are working on tailoring our algorithm to be able to analyse this type of data, as well as investigating the potential of running the TCCON pipeline (GGG2020) on spectra from telluric standard stars. On the COCCON side, we are in the early talking stages of potentially temporarily placing a COCCON spectrometer in an astronomical observatory, which would provide a robust way to validate the results from our algorithm.